# Dual-Comb Ghost Imaging with Transformer-Based Reconstruction for Optical Fiber Endomicroscopy

**David Dang**[1,3]    **Myoung-Gyun Suh**[2]    **Maodong Gao**[2]    **Byoung Jun Park**[2]
**Beyonce Hu**[1]    **Yucheng Jin**[1]    **Wilton J.M. Kort-Kamp**[3]    **Ho Wai (Howard) Lee**[1]

[1]University of California, Irvine    [2]NTT Physics and Informatics Laboratories
[3]Los Alamos National Laboratory
{dangd5,beyonceh,yuchej9,Howardhw.lee}@uci.edu
{myoung-gyun.suh,byoungjun.park}@ntt-research.com, mgao@caltech.edu
kortkamp@lanl.gov

## Abstract

Endoscopic imaging is indispensable for visualizing internal organs, yet conventional systems remain bulky and costly because they rely on large, multi-element optics, which limits their ability to access and image certain areas of the body. Achieving high-quality endomicroscopy with hundred micron-scale and inexpensive hardware remains a grand challenge. Optical fibers offer a sub-millimeter-scale imaging conduit that could meet this need, but existing fiber-based approaches typically require either raster scanning or multicore bundles, which limit resolution and speed of imaging. In this work, we overcome these limitations by combining dual-comb interferometry with optical ghost imaging and advanced algorithm. Optical frequency combs enable precise and parallel speckle illumination via wavelength-division multiplexing through a single-core fiber, while our dual-comb compressive ghost imaging approach enables snapshot detection of bucket-sum signals using a single-pixel detector, eliminating the need for both spatial and spectral scanning. To reconstruct images from these highly compressed measurements, we introduce **Optical Ghost-GPT**, a transformer-based image reconstruction model that enables fast, high-fidelity recovery at low sampling ratios. Our dual-comb ghost imaging approach, combined with the novel algorithm, outperforms classical ghost imaging techniques in both speed and accuracy, enabling real-time, high-resolution endoscopic imaging with a significantly reduced device footprint. This advancement paves the way for non-invasive, high-resolution, low-cost endomicroscopy and other sensing applications constrained by hardware size and complexity.

## 1 Introduction

Endoscopes are an important tool in modern medical diagnostics, enabling direct visualization of internal organs and tissues. From gastrointestinal investigations to bronchoscopies and laparoscopies, endoscopic techniques are widely used for both diagnostic and therapeutic purposes. Traditional endoscopes typically consist of a long, flexible or rigid tube equipped with a light source and a camera system to capture real-time images [1]. Despite their essential role in clinical practice, conventional endoscopes face several critical challenges: **(a)** The complex electronics and imaging systems increases an endoscope's size to the order of millimeters to centimeters— resulting in invasive and uncomfortable procedures on a patient; **(b)** Moreover, the imaging capabilities are typically limited by the size of the optics and cameras and prevent the imaging of small body parts; **(c)** Additionally, these imaging systems play a role in the large cost of endoscopes, with average prices in the range of several tens of thousands of dollars [2].

39th Conference on Neural Information Processing Systems (NeurIPS 2025).

Fiber endomicroscopy offers a markedly smaller alternative. By exploiting single-core, multicore, or multimode optical fibers with outer diameters below 500 micron, it is possible to deliver light and collect signals through a conduit scarcely thicker than a human hair. Early demonstrations, including confocal[3] and multiphoton fluorescence microendoscopes[4], established that cellular-scale imaging could be achieved with a sub-millimeter footprint. Subsequent advances such as scanning-fiber endoscopes and microelectromechanical (MEMS) scanners improved field of view[5] , while computational approaches have enabled lensless, holographic, and light-field reconstructions through flexible probes as thin as 200 micron[6, 7]. Despite these successes, most fiber-based systems still require either mechanical components such as MEMS scanner or rotating torque coil for raster scanning or coherent fiber bundles[8] whose inter-core spacing limits resolution, operation speed, and reduces fill factor.

Ghost imaging provides a way to bypass these bottlenecks by replacing pixelated cameras with correlated intensity measurements from a single-pixel (bucket) detector[9]. In a typical implementation, a sequence of known illumination patterns are projected onto the sample, and statistical correlations between these patterns and the measured total intensities are used to computationally reconstruct the image. Ghost imaging has demonstrated advantages in low light conditions due to improved signal-to-noise ratios[10, 11, 12], the ability to image through scattering media[13, 14], and compatibility with single-pixel detectors - making it well suited for fiber-based imaging, where 2D detector arrays are impractical[15, 16]. However, classical ghost imaging remains slow because each pattern must be projected sequentially, and iterative reconstruction algorithms often converge slowly, are prone to low image fidelity, or stall at low sampling ratios.

In this work, we combine dual optical frequency comb (dual-comb, for short) interferometry[17, 18, 19] with compressive ghost imaging to realize snapshot speckle imaging through a single-core fiber, eliminating the need for slow spatial or spectral scanning while preserving a minimal footprint. For image recovery, we introduce **Optical Ghost-GPT**, a transformer-based reconstruction model that enables real-time, high-fidelity imaging. Our contributions include: **(1)** First demonstration of optical fiber-based ghost imaging using a hardware-software co-design that combines dual-comb interferometry and deep learning, **(2)** superior reconstruction speed and resolution through the hardware-software co-design approach, **(3)** a robust transformer-based framework that maintains performance in noisy environments.

## 2   Background on Single-Pixel and Ghost Imaging

The most straightforward form of single-pixel imaging involves raster scanning, where a single-pixel detector sequentially measures light intensity at each pixel location, requiring $N^2$ measurements for an $N \times N$ image. In contrast, *ghost imaging* enables image reconstruction from significantly fewer measurements by illuminating the object (x) with structured light patterns (A) and using a single-pixel detector to measure the total transmitted or reflected intensity [20, 21].

The *sampling ratio* is defined as:

$$\beta = \frac{M}{N^2}, \tag{1}$$

where $M$ is the number of structured light patterns used. Each projected pattern $(A^{(m)})$ is modulated by the object and measured as a scalar intensity, commonly referred to as the bucket sum:

$$b^{(m)} = \sum_{i=1}^{N} \sum_{j=1}^{N} A_{i,j}^{(m)} \cdot x_{i,j}. \tag{2}$$

Early methods, such as Differential Ghost Imaging (DGI), used the bucket detector signals to compute weighted sums of the illumination patterns for object reconstruction, but these approaches often produced low-fidelity results  [22]. More advanced methods recast ghost imaging as a linear system:

$$\mathbf{b} = \Psi \mathbf{x}, \tag{3}$$

where $\Psi$ is the sensing matrix formed by flattening and stacking the illumination patterns, and $\mathbf{x}$ is the vectorized image. In this form, a standard solution using the Moore–Penrose pseudoinverse (PI) is given by:

$$\mathbf{x} = \Psi^{\dagger} \mathbf{b}. \tag{4}$$

To enhance reconstruction quality, compressed sensing techniques leverage transform-domain sparsity [23, 24, 25], often employing $\ell_1$ or $\ell_2$ regularization, resulting in

$$\mathbf{b} = \Psi\Phi\boldsymbol{\alpha}, \quad \text{with } \mathbf{x} = \Phi\boldsymbol{\alpha}. \tag{5}$$

Here, $\Phi$ is a sparsifying basis (e.g., DCT or wavelets), and $\boldsymbol{\alpha}$ is a sparse coefficient vector. Solutions are obtained using iterative optimization methods such as Iterative Hard Thresholding (IHT) [23], Fast Iterative Shrinkage-Thresholding Algorithm (FISTA) [24], or Alternating Direction Method of Multipliers (ADMM) [25]. Definitions of the variables used above are provided in Appendix A.

Deep neural networks and generative models have been used to improve image reconstruction fidelity [26]. One common approach first applies a classical reconstruction algorithm to produce a low-quality image, which is then enhanced using a trained neural network—such as a CNN, U-Net, or, more recently, a diffusion model [27] [28, 29]. This method benefits from the ability to incorporate learned priors for denoising and super-resolution. However, its performance depends on the quality as well as speed of the initial reconstruction and requires large datasets of paired examples. Another approach uses an end-to-end strategy by embedding elements of the ghost imaging process directly into the neural network architecture [30, 31]. This method is typically faster, as it bypasses classical reconstruction algorithms entirely. However, prior studies have been constrained to low-resolution images (e.g., 28×28) and relied on binary mask patterns.

A critical oversight in most work on ghost imaging is the assumption of simultaneous pattern acquisition. This ignores the slow, serial nature of real-world SLM/DMD-based systems, which makes dynamic imaging impractical. In contrast, our hardware-software co-design uses dual-comb interferometry to achieve true parallel acquisition. This makes high-fidelity imaging of fast-moving objects feasible and aligns the physical system with the assumptions of modern reconstruction algorithms.

## 3    Dual-Comb Ghost Imaging: Experimental Details

Figure 1 illustrates the concept of dual-comb ghost imaging. An optical frequency comb is a laser source whose spectrum consists of a series of equally space, mutually coherent narrow frequency lines, resembling the teeth of a comb. In our approach, a set of 2D speckle patterns $H$, comprising uncorrelated speckle distributions across different comb line frequencies, is generated at the fiber tip using dual optical frequency combs (OFCs) and projected onto the 2D target object $x$. The encoded intensity distribution ($H \times x$) is detected by a single-pixel photodetector, producing a dual-comb interferogram. A fast Fourier transform (FFT) converts this time-domain signal into frequency-domain bucket-sum data $y = H \times x$. By multiplexing light through the optical fiber, we parallelize the speckle projection process in ghost imaging, significantly increasing imaging speed.

For the dual-comb OFC source, we use two electro-optic (EO) combs with slightly different free spectral ranges ($f_{FSR}$). A continuous-wave (CW) fiber laser at $1550\,\mathrm{nm}$ is first amplified and split into two beams via a 50/50 fiber coupler. Each beam is frequency-shifted using an acousto-optic modulator, introducing a center frequency offset $\Delta f_{center}$. These beams are then independently modulated by resonant EO modulators driven at $f_{FSR}$ and $f_{FSR} + \Delta f_{FSR}$, respectively, where $f_{FSR} = 20\,\mathrm{GHz}$ and $\Delta f_{FSR} = 200\,\mathrm{Hz}$. The resulting EO combs are recombined using a 50/50 fiber coupler and amplified to compensate for insertion losses. When two mutually coherent combs with slightly different repetition rates are combined and photodetected, their interference produces a set of beat signals in the radio-frequency (RF) domain. This allows fast, precise electronic detection without slow optical spectrum analyzers, while mutual coherence enables coherent averaging for improved signal-to-noise ratio (SNR) [17].

Before the free-space imaging setup (see Figure 1b), a Waveshaper is placed in the system to filter individual comb lines if needed. For bucket-sum measurements, the Waveshaper transmits the full dual-comb spectrum. The multimode fiber used in the experiments has a core diameter of 200 um and exhibits hundreds of core modes, resulting in speckle patterns when light is coupled to the core of the optical fiber. The generated speckle patterns are then collimated and passed through the target object. A 50/50 beam splitter divides the beam into signal and reference paths, allowing simultaneous acquisition and suppression of common-mode temporal noise. Calibration measurements are performed without the target to correct for optical path imbalances. The set of speckle patterns ($H$) without the target object is separately recorded using a 2D camera by selecting individual comb lines with the Waveshaper, either before or after the imaging. The imaging target is a

negative USAF 1951 resolution chart mounted on a motorized stage. Bucket-sum signals are acquired using 500 kHz bandwidth free-space InGaAs photodetectors, and speckle patterns are recorded with a $256 \times 256$ pixel InGaAs camera.

The imaging experiment uses approximately 200 comb lines spanning from $1530$ nm to $1565$ nm. The speckle patterns at different frequencies are uncorrelated, as indicated by low Pearson correlation coefficients (Figure 2b), with slight residual correlations attributed to background contributions at the pattern edges. Power variations among comb lines are normalized during processing. Figure 2c shows the signal and reference RF combs obtained via FFT of the interferograms. The amplitude ratios of corresponding comb peaks represent the bucket-sum signals that encode the image information. A calibration measurement without the target is used to correct for the path differences.

For image reconstruction, we used the calibrated bucket-sum signals and the measured speckle patterns. Figure 2d shows the calibrated measurement compared to the theoretical predictions, showing excellent agreement. Images are initially reconstructed using the Moore-Penrose pseudoinverse, recovering the target image even at a $0.3\%$ sampling ratio. However, background noise from speckle structures remains significant. To address this, we developed Optical Ghost-GPT, a transformer-based reconstruction algorithm that substantially suppresses noise and enhances image quality. Details of Optical Ghost-GPT will be discussed in the following section.

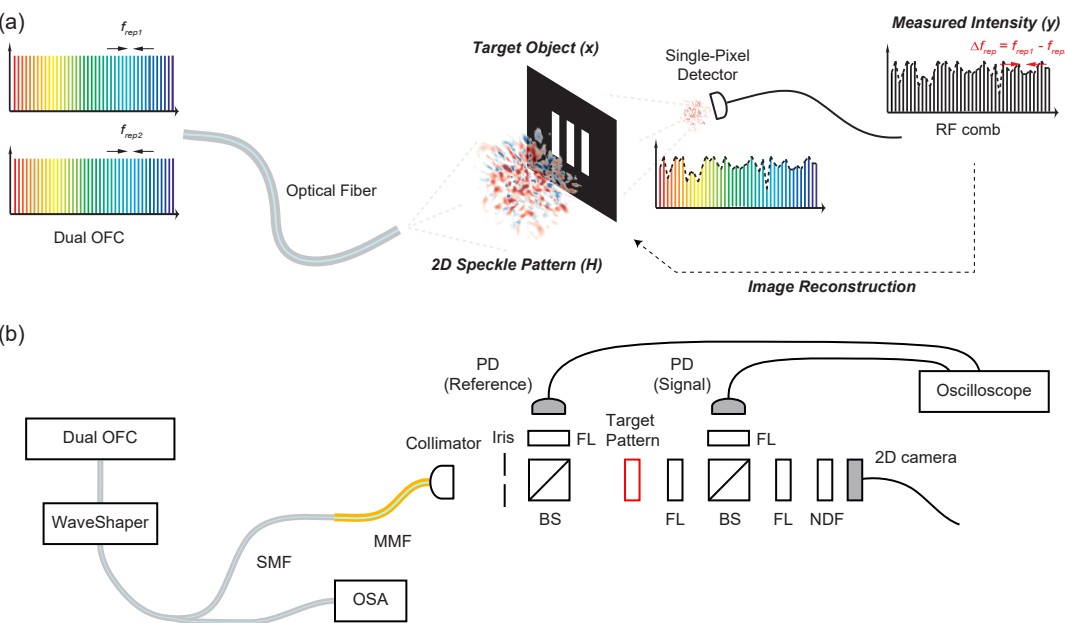

Figure 1: **Optical Fiber-based Dual-Comb Ghost Imaging.** (a) Conceptual illustration of optical fiber-based ghost imaging using dual optical frequency combs. Each comb line generates an uncorrelated speckle pattern at the fiber tip, and the set of speckle patterns is mapped onto the target object $x$. The encoded speckle pattern ($H \times x$) is collected by a single-pixel detector, and the resulting dual-comb interference signal ("interferogram") is recorded. The time-domain interferogram is converted into the frequency domain by fast Fourier transform (FFT), yielding the bucket-sum information ($y = H \times x$). The dual-comb technique allows for snapshot ghost imaging, providing a highly parallel, high-SNR, and fast imaging capability. Notably, the use of comb-based wavelength-division multiplexing (WDM) and bucket-sum compressive imaging reduces the physical dimensions of both input and output hardware to essentially zero-dimensional (i.e., single spot or pixel), making the system particularly suitable for applications requiring compact form factors, such as endomicroscopy. (b) Experimental setup diagram. SMF: Single Mode Fiber, OSA: Optical Spectrum Analyzer, PD: Photodetector, FL: Focusing Lens, BS: Beam Splitter, NDF: Neutral Density Filter.

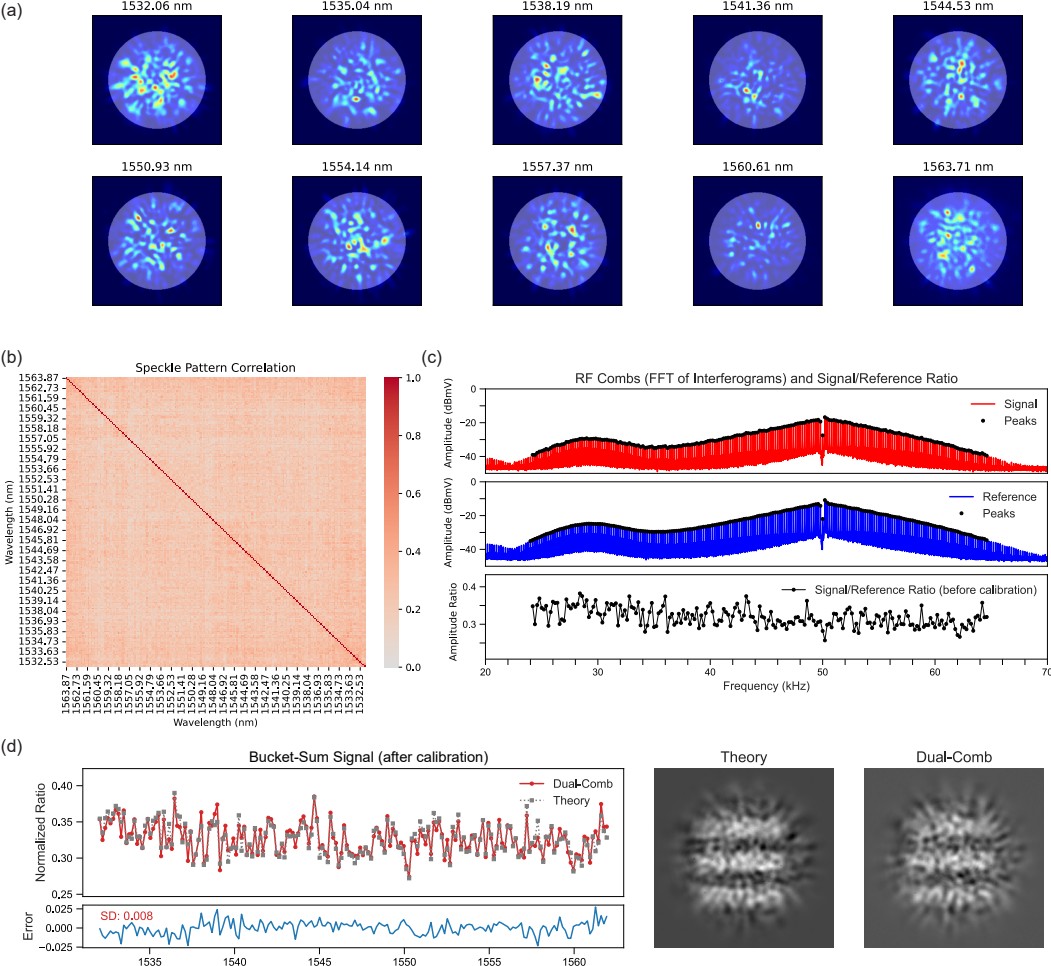

Figure 2: **Experimental Details.** (a) Speckle patterns measured at ten different comb line frequencies. Each image contains $256 \times 256$ pixels. (b) Pearson correlation coefficients calculated between speckle patterns across the wavelength range from 1532 nm to 1564 nm. The correlation is computed within the bright circular region of each speckle pattern. The low off-diagonal values indicate that the speckle patterns are largely uncorrelated. Residual correlations are attributed primarily to the common background near the edges of the speckle patterns. (c) Radio-frequency (RF) combs generated from dual-comb interference signals measured by the signal and reference single-pixel detectors (upper and middle panels). The ratio between the comb peak amplitudes of the signal and reference RF combs is shown (bottom panel). Differences between the signal and reference beam paths are calibrated using measurements performed with and without the target object. (d) The calibrated bucket-sum signal measured from the dual-comb experiment shows excellent agreement with the theoretical prediction, which is obtained by masking the ground-truth pattern onto the stored speckle pattern images. The standard deviation (SD) between the two bucket-sum signals is 0.008. Image reconstruction is performed using the Moore-Penrose pseudoinverse algorithm at a sampling ratio (SR) of 0.29%. Both the theoretical reconstruction (left) and the experimental dual-comb reconstruction (right) successfully recover the ground-truth patterns.

# 4 Transformer Modeling

Transformers are a powerful deep learning architecture originally introduced for natural language processing (NLP) tasks but have since found applications in various domains [32], including computer vision [33], speech processing [34], and scientific data analysis [35]. They leverage a mechanism called self-attention to model long-range dependencies and capture contextual relationships within sequences [36]. Unlike traditional recurrent or convolutional networks, transformers process entire sequences simultaneously, making them highly efficient for parallel computation. Their ability to learn complex patterns from large datasets has made them the backbone of state-of-the-art models like BERT, GPT [37], and Vision Transformers (ViTs) [38] [39].

In typical ViTs, the image is broken up into equally sized patches, which serve as the ViTs' token. In order to adapt this methodology to dual-comb ghost imaging, we propose concatenating the flattened illumination patterns that make the sensing matrix, $\Psi$, with the bucket value to form the token for our model.

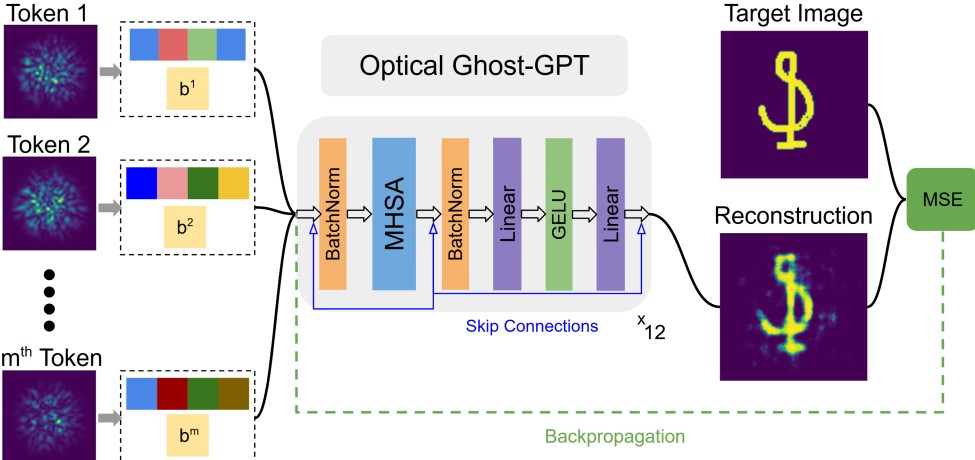

Figure 3: **Schematic of Optical Ghost-GPT**. The speckle patterns are first compressed into a latent space and then concatenated with the corresponding bucket measurements to form the input token sequence.

**Model Architecture.** We introduce **Optical Ghost-GPT** (or simply **Ghost-GPT**), a transformer-based model designed for structured image reconstruction from ghost imaging measurements. The model leverages stacked self-attention mechanisms and residual feedforward blocks to model long-range dependencies across the contextual input.

**Input Embedding.** The model receives two primary inputs in each token: a flattened image of the speckle pattern $\boldsymbol{\Psi}^m \in \mathbb{R}^N$, where $N = 256 \times 256$, and bucket sum value $\mathbf{b^m}$. The image vector is projected via a learnable linear transformation to a latent embedding of dimension `embedding_dim` $-$ $1$, and afterwards, the latent representation of the image vector and its corresponding bucket sum are concatenated. To encode positional structure, we add learned positional embeddings of size `embedding_dim` to each token in the sequence:

$$\mathbf{z}_i = \mathbf{e}_i + \mathbf{p}_i, \quad i = 1, \ldots, C,$$

where $\mathbf{e}_i$ is the input token embedding, $\mathbf{p}_i$ is the corresponding positional embedding, and C=250 is the context size determined by the theoretical maximum number of RF comblines in our setup.

**Transformer Blocks.** The architecture contains a stack of $L = 12$ transformer blocks, each composed of a multi-head self-attention (MHSA) mechanism and a two-layer feedforward network. The attention mechanism employs $H$ number of heads, computed as:

$$\text{Attention}(Q, K, V) = \text{Softmax}\left(\frac{QK^\top}{\sqrt{d_k}}\right)V,$$

where $Q$, $K$, and $V$ are linear projections of the input sequence, and $d_k = \texttt{embedding\_dim}$. We also apply a dropout mask with a value of 0.1 to ensure robustness of the model and allow the model to generalize to missing shots and buckets. Each block includes residual connections and a batch normalization operation placed before and after the MHSA layer, defined as:

$$\text{BatchNorm}(\mathbf{x}) = \frac{\mathbf{x} - \mu}{\sqrt{\sigma^2 + \varepsilon}}, \quad \mu = \mathbb{E}[\mathbf{x}], \quad \sigma^2 = \text{Var}[\mathbf{x}].$$

The feedforward network consists of two linear transformations with a gaussian error linear unit (GELU) activation; A final transformer block is appended after the main stack to further refine the sequence representation.

$$\text{FFN}(\mathbf{x}) = \text{Linear}_2\left(\text{GELU}(\text{Linear}_1(\mathbf{x}))\right).$$

**Output Projection.** The final token representations are projected to $\mathbb{R}^{16}$ via a linear layer and the output is flattened to a tensor of size $(C \times 16)$ and passed through a final linear layer to reconstruct the original image vector in $\mathbb{R}^{256 \times 256}$. A sigmoid activation is applied to constrain the output to the range $[0, 1]$, consistent with normalized image intensities:

$$\hat{\mathbf{x}} = \sigma\left(\text{Linear}_{\text{final}}(\text{Flatten}(\mathbf{z}))\right).$$

**Network Training.** We first obtain a set of speckle patterns from the 2-D camera during the calibration phase of the experiment. (We emphasize that the 2-D camera is only needed to obtain the initial speckle patterns and can be removed during imaging). For this experiment, we obtained 188 speckle patterns, which are then used to generate synthetic buckets sums via a convolution between the digitized speckle pattern and images from the MNIST and OMNIglot datasets. We form our labeled dataset of synthetic bucket sums as the x-label and its corresponding target images as the y-label. (See Appendix B for train/test split and computational resources used).

In training, Ghost-GPT predicts the target image, given the speckle pattern and bucket sums as the input. We use mean squared error as our loss function and an AdamW optimizer with a learning rate of 0.0003 and a weight decay of 0.001. For the following reconstruction results, we used a model with the number of attention heads set to 8 and the $\texttt{embedding\_dim}$ set to 32 based on a hyperparameter sweep. (See Appendix C for further information on the hyperparameter analysis).

## 5  Ghost-GPT Reconstruction Results

In this section, we examine the reconstruction results in simulation using our experimental speckle pattern results. Figure 4 shows a series of reconstruction compared with their true images.

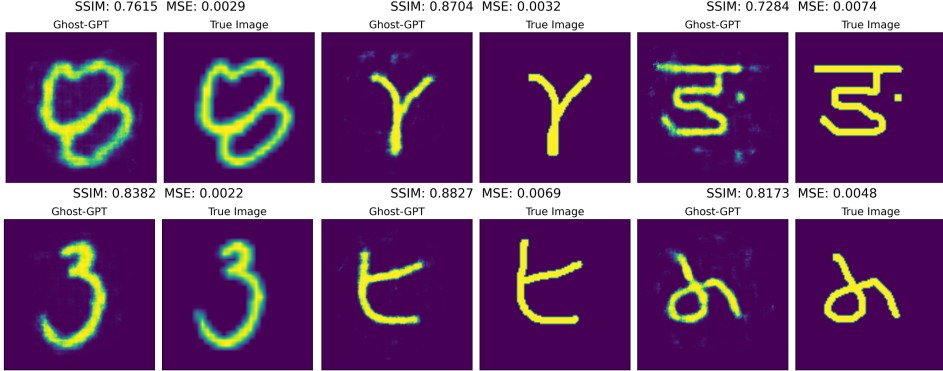

Figure 4: Reconstructions from Ghost-GPT versus the true image with the Structural Similarity Index Measure (SSIM) and Mean Squared Error (MSE) displayed above.

We emphasize that we achieve very high SSIM (greater than 0.7) and low MSE (less than 0.02) values despite having a sampling ratio of only 0.29%. We observe that the model is able to preserve fine structural details and clear object boundaries, highlighting the model's ability to recover meaningful

image content from minimal measurements. However, the reconstructions can exhibit variations in intensity across the object, especially in images with long, thin structures or uniform intensity profiles. These artifacts likely stem from the inherent non-uniformity of the speckle patterns generated by the multi-mode fiber. This issue could potentially be mitigated by employing a more uniform speckle distribution or by including a smoothness term, such as total variational loss, in the loss function.

We also compared our model against classical reconstruction algorithms on 256 images from our validation dataset. (256 images were chosen due to the long reconstruction times associated with the iterative FISTA algorithm). Table 1 compares the MSE and SSIM of previously discussed reconstruction algorithms- as expected the simpler reconstruction algorithms such as Differential Ghost Imaging and the Moore-Penrose Psuedo Inverse perform worse than compressed sensing methods like FISTA. However, Ghost-GPT outperforms these classical algorithms giving an average **MSE of 0.008** and **SSIM of 0.788**, while being approximately **263x** faster than FISTA. (We set $\epsilon =$ 50 after performing a hyperparameter sweep and choosing the best performing SSIM). The extremely fast reconstruction speed of **14 ms** enables **real-time, video frame-rate ghost imaging in optical fibers**. Importantly, while the computational reconstruction speed can be further improved with better computing hardware, the fundamental limit of image reconstruction is set by the repetition rate difference of the dual-comb, which is typically a few hundred Hz to several kHz in our experiments. With a larger repetition rate difference, a much higher frame rate is possible. However, this requires high-bandwidth photodetection, and the trade-off between the sampling ratio and frame rate must be considered in accordance with the Nyquist condition.

Table 1: Classical Algorithms vs Ghost-GPT

| Algorithm | MSE | SSIM | Computational Time (ms) |
|---|---|---|---|
| Differential Ghost Imaging | 0.184 ($\sigma$=0.037) | 0.042 ($\sigma$=0.022) | 15.7 ($\sigma$=0.138) GPU |
| Moore–Penrose Pseudo-Inverse | 0.093 ($\sigma$=0.027) | 0.055 ($\sigma$=0.027) | 5640 ($\sigma$=52.1) CPU
157 ($\sigma$=3.27) GPU |
| FISTA (200 iters, $\epsilon$=50) | 0.045 ($\sigma$=0.017) | 0.092 ($\sigma$=0.019) | 3680 ($\sigma$=125) CPU
10500 ($\sigma$=93.8) GPU |
| **Ghost-GPT (Ours)** | **0.008 ($\sigma$=0.009)** | **0.788 ($\sigma$=0.076)** | **14.0 ($\sigma$=0.457) GPU** |

With our experimental setup, we collected physical intensity measurements of a USAF resolution test target (ThorLabs R3L3S1N 3" x 3"). Figure 5 and Table 2 shows the reconstruction results of a stripe (top) and the number 2 (bottom) compared with classical algorithms and Ghost-GPT and the corresponding MSE and SSIM values. In real world environments, we verify that Ghost-GPT is able to successfully reconstruct images with higher fidelity for both targets compared to classical methods.

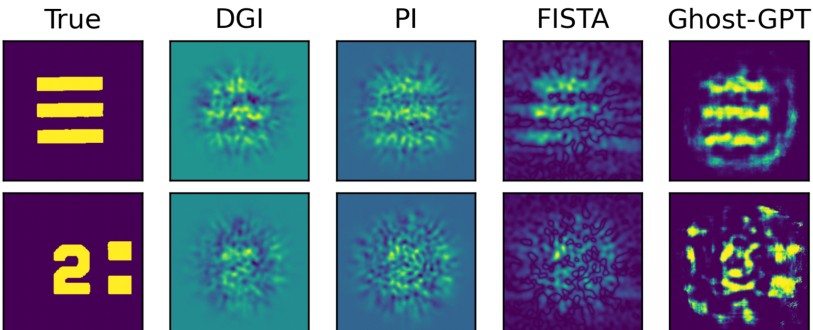

Figure 5: Reconstruction using experimental bucket measurements compared with classical algorithms and Ghost-GPT. (Top row): Striped lines correspond to group 0, element 4 (1.41 line pairs per millimeter). (Bottom Row): The number 2 corresponds to group 0, element 2 (1.12 lp/mm).

## 6   Robustness of Ghost-GPT in Experimental Imaging

To examine the effect of noise in our experimental measurement, we performed a signal-to-noise ratio analysis by adding varying amounts of noise to simulated buckets and gauged the quality

Table 2: MSE/SSIM for Experimental Targets with Classical Algorithms and Ghost-GPT

| Experimental Target | DGI | PI | FISTA | Ghost-GPT (Ours) |
|---|---|---|---|---|
| Stripes | 0.231/0.030 | 0.140/0.028 | 0.072/0.057 | **0.058/0.470** |
| Number 2 | 0.204/0.036 | 0.138/0.028 | 0.084/0.064 | **0.077/0.484** |

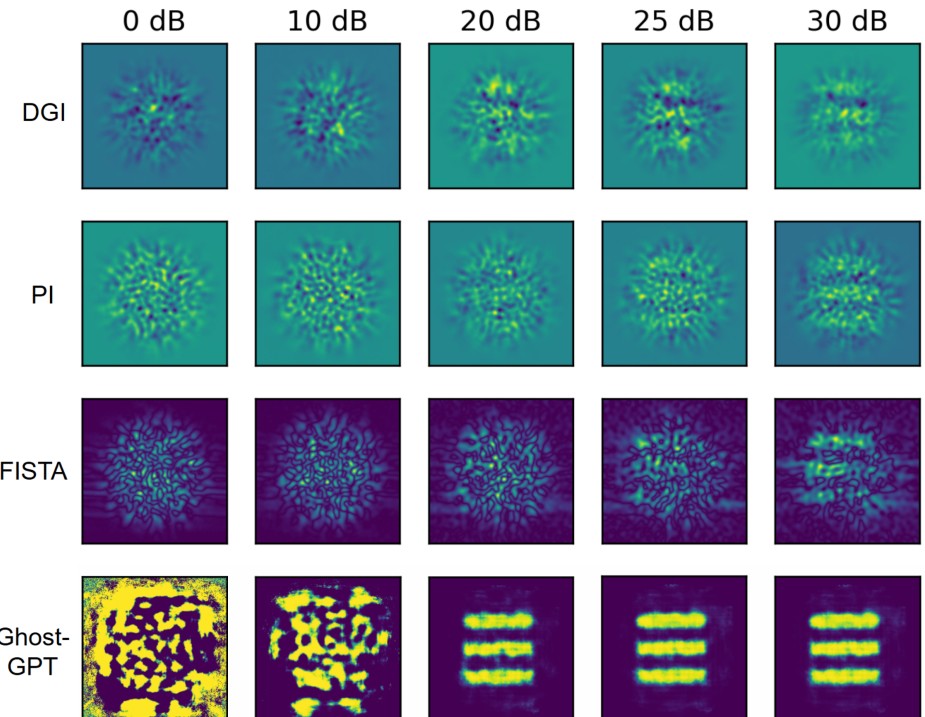

Figure 6: Simulated reconstructions of the USAF striped targets with artificial noise added to the buckets. The signal to noise ratio in dB is shown label at the top with the corresponding reconstruction algorithms on the left-hand side.

of image reconstructions (Figure 7). The artificial noise added to the buckets were calculated by sampling points from a gaussian with mean of zero and standard deviation of one and scaling it inversely proportional to the value of the SNR. (See Appendix D.1 for more information about the SNR calculation). Ghost-GPT demonstrates strong robustness to noise, outperforming classical reconstruction algorithms, with recognizable reconstructions achievable at SNR levels above 20 dB for Ghost-GPT. (See Appendix Figure 9 for plots of the calculated MSE and SSIM values vs SNR). Based on this analysis, our experimental bucket measurements are approximately equivalent to 20-25 dB. For further experimental results, code/data availability, and discussion about project limitations, please refer to D.2, E, and G.

# 7   Comparison with Other Deep Learning Models

We compared Ghost-GPT, our transformer-based reconstruction model, against two deep learning baselines: a decoder-only CNN and a U-Net with skip connections. All models were matched in parameter count ($\sim$ 270M) and trained on identical speckle–bucket inputs for fair comparison. As shown in Table 3, Ghost-GPT achieves the lowest mean-squared error (MSE = 0.0068), representing a 47.7% reduction vs. CNN and 38.18% vs. U-Net across the entire validation dataset, while maintaining comparable structural similarity (SSIM = 0.787).

Training dynamics further show that Ghost-GPT converges to a lower validation loss within the same number of epochs, supporting the suitability of the transformer's self-attention mechanism

Table 3: Image Reconstruction Quality on Various Neural Network Architectures

| Model Architecture | MSE | SSIM | Computational Time (ms) |
|---|---|---|---|
| **Ghost-GPT (Ours)** | **0.0068 ($\sigma$=0.0077)** | 0.787 ($\sigma$=0.073) | 14.0 ($\sigma$=0.457) (GPU) |
| CNN | 0.013 ($\sigma$=0.0086) | 0.8079 ($\sigma$=0.0395) | **6.26 ($\sigma$=0.41) (GPU)** |
| UNet | 0.011 ($\sigma$=0.0073) | **0.8316 ($\sigma$=0.0696)** | 15.09 ($\sigma$=0.032) (GPU) |

for modeling the global correlations inherent in ghost imaging measurements. These findings align with recent studies reporting transformer models outperforming conventional deep architectures in computational imaging tasks [40]. While our model was trained using MSE loss for quantitative comparability, the observed SSIM gap suggests that training with a perceptual loss or SSIM-weighted objective could further enhance structural fidelity without sacrificing reconstruction accuracy. We leave this as a promising direction for improving perceptual quality in future iterations of Ghost-GPT. It is worth noting that, with simple binary datasets, it was difficult to compare GhostGPT to other deep-learning models because the performance metrics gave mixed signals within a narrow range. With a larger number of speckle patterns and more realistic grayscale datasets, we should be able to conduct a more systematic performance comparison with other deep-learning models.

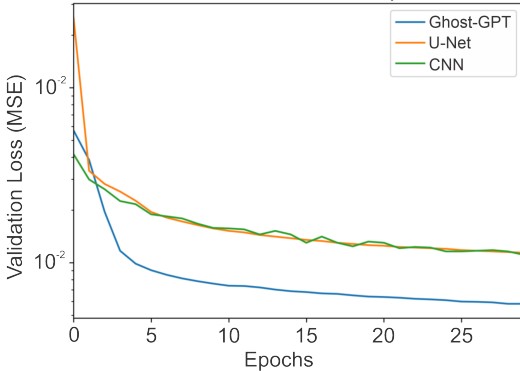

Figure 7: Loss history of different deep learning models compared with Ghost-GPT. Our transformer-based model consistently outperforms both U-Net and CNN for large majority of epochs.

## 8 Conclusion

In this work, we demonstrate a novel dual-comb ghost imaging using only a single-core fiber and a single-pixel detector by combining dual-comb interferometry with a Transformer tailored for fast, high-quality reconstruction—a hardware–software co-design. Our GPT-like deep learning model is application-aware: paired tokens jointly encode each speckle pattern and its corresponding bucket sum to meet real-time, high-fidelity goals. Leveraging such deep learning architectures, we achieved high-fidelity reconstructions (average SSIM of 0.788) at ultra-low sampling ratios (as low as 0.29%), significantly outperforming classical methods in both resolution and efficiency. In our first experimental demonstration, the number of uncorrelated speckle patterns was limited to about 200, so we used binary datasets suitable for reconstruction at such low sampling ratios. This, however, is a prototype-stage constraint, not a fundamental limitation of the method. Realistic grayscale reconstructions could be enabled by increasing the number of comb lines, which can be achieved by broadening the spectrum and reducing speckle correlation width. Future work will focus on achieving video-rate grayscale imaging, enhancing the model via improved sensing matrix design and noise-aware training, and developing efficient beam collection in reflective-mode setups for practical applications. This framework holds strong potential for emerging applications in biomedical imaging and imaging within extreme, low-light environments. The ability to reconstruct high-resolution images from sparse, low-light measurements makes it particularly suited for imaging live or light-sensitive biological samples, for real-time, minimally invasive procedures such as fiber-based endoscopy, or dynamic tissue monitoring—offering a promising pathway toward next-generation medical diagnostics.

## Acknowledgments and Disclosure of Funding

DD, MGS, and HWHL conceived the concept of Optical Ghost-GPT. DD and WJMKK designed and implemented the GPT model architecture. DD, BH, and MGS carried out model training, evaluation, and comparative analysis with classical reconstruction algorithms. MGS, MG, and BJP led the experimental integration of dual-comb spectroscopy with optical fiber-based ghost imaging. YJ explored various optical fiber types for this application and conducted speckle pattern simulations using Finite Difference Time Domain (FDTD) methods. All authors contributed to the discussions and writing of the manuscript.

DD acknowledges the support by UCI-LANL-SoCal Hub graduate fellowship program. WJMKK acknowledges the LANL Laboratory Directed Research and Development program for funding under project 20250492ER.

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

# A  Variable Definitions

The equations in the main text use the following symbols:

- $A^{(m)} \in \mathbb{R}^{N \times N}$: The $m^{\text{th}}$ structured illumination pattern,
- $x \in \mathbb{R}^{N \times N}$: The unknown image to be reconstructed,
- $b^{(m)}$: Total detected intensity for the $m^{\text{th}}$ pattern,
- $\mathbf{b} \in \mathbb{R}^{M \times 1}$: Vector of all scalar intensity measurements,
- $\Psi \in \mathbb{R}^{M \times N^2}$: Sensing matrix composed of flattened patterns,
- $\mathbf{x} \in \mathbb{R}^{N^2 \times 1}$: Flattened version of the image $x$,
- $\Phi \in \mathbb{R}^{N^2 \times N^2}$: Sparsifying basis (e.g., DCT, wavelets),
- $\boldsymbol{\alpha} \in \mathbb{R}^{N^2 \times 1}$: Sparse coefficients in the transform domain.

# B  Dataset Generation, Train and Test Split

Since the bucket signal is effectively a convolution between the object and each speckle pattern, it is computationally efficient to synthetically generate labeled training data by pairing known objects with their corresponding bucket sums. For this, we use open-source image datasets and simulate the measurement process. Our dataset includes 19,280 from OMNIglot and 19,280 images from MNIST with a train/val split of 33,000/5,560 respectively. All images are resized to 256 × 256 to match the resolution of the recorded speckle patterns. We use a batch size of 32 for the training dataset and a batch size of 64 for the validation dataset.

## B.1  Licensing

**MNIST:**  Originally published by [41], downloaded via `torchvision.datasets.MNIST`, and licensed under the *Creative Commons Attribution-Share Alike 3.0* license (see `http://yann.lecun.com/exdb/mnist/`).

**Omniglot:**  Originally published in [42], downloaded via `torchvision.datasets.Omniglot`, and licensed under the *MIT License* (see `https://github.com/brendenlake/omniglot`).

# C  Hyperparameter Analysis

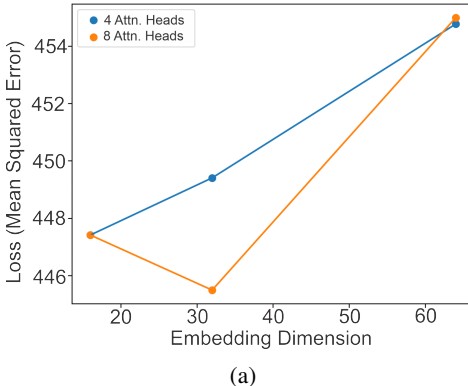
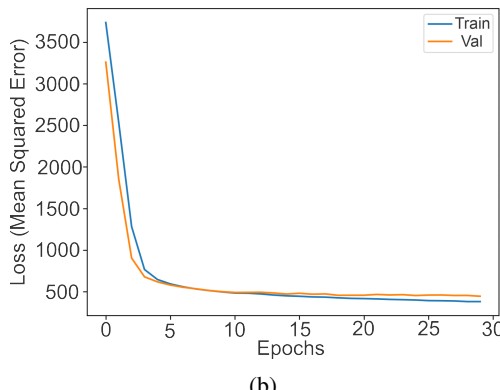

(a)                                                    (b)

Figure 8: The lowest loss value on the validation dataset as a function of the number of attention heads and embedding dimensions (a) comparison with pattern embedding (b) log scale loss values of models with and without pattern embedding

For hyperparameter tuning, we performed a simple parameter sweep by changing the number of attention heads at values of 4 and 8 as well as adjusting the embedding dimension at values of 16, 32,

and 64. Due to memory constraints, we limited our parameter search to these values. As previously stated, we use mean squared error as our loss function and an AdamW optimizer with a learning rate of 0.0003 and a weight decay of 0.001. In our loss function, we add the error in each pixel and then average over the batch size.

Over 30 epochs, the model took approximately 1 hour to train on a Linux workstation, using an AMD Ryzen Threadripper 3990X 64-Core Processor and four NVIDIA A6000 GPUs. Our chosen model (8 attention heads, embedding dimension 32) contained $\sim$ 270 million trainable parameters and demonstrates convergence after 10 epochs (Appendix Figure 8b)- displaying no characteristics of overfitting.

## D   Further Experimental Results

### D.1   Different Algorithms vs SNR

The SNR was calculated via the following equation:

$$\mathbf{b}_{noisy} = \mathcal{N}(0,1) * \sigma + \mathbf{b}, \quad \sigma = \frac{\mathbb{E}[\mathbf{b}]}{\text{SNR}^{10}}$$

We sample points from a gaussian of mean 0 and std 1, which is scaled by $\sigma$, the standard deviation. $\sigma$ is calculated the by averaging over the collection of bucket sum and then dividing by the SNR value raised to the $10^{\text{th}}$ power. The true bucket sum, $\mathbf{b}$, is added to the previously calculated quantity to create the noisy bucket, $\mathbf{b}_{\text{noisy}}$. Previous works in ghost imaging formulated their SNR analysis using these exact definitions.

Appendix Figure 9 plots the MSE and SSIM of the reconstructed images compared with classical reconstruction algorithms and Ghost-GPT.

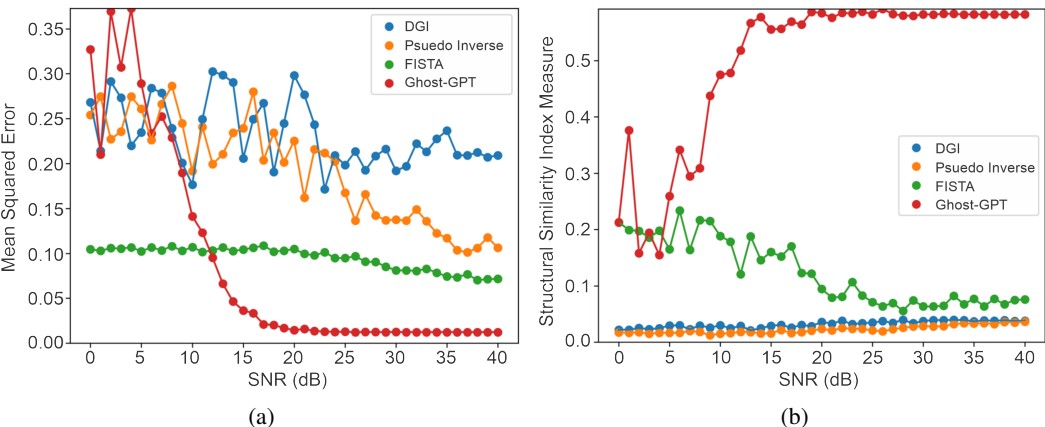

(a)                                                (b)

Figure 9: SNR ratio in the buckets versus (a) MSE and (b) SSIM of the USAF target

### D.2   Resolution Analysis

In ghost imaging, resolution is fundamentally linked to the spatial characteristics of the light patterns used to probe the object—specifically, the grain size of these patterns. Grain size refers to the typical scale or correlation length of the intensity fluctuations in the illumination patterns (often speckle patterns). This grain size determines the finest detail that can be distinguished in the reconstructed image.

In this section, we evaluated the smallest resolvable feature size achievable by Ghost-GPT in simulation using our current experimental speckle pattern set (Appendix Figure 10. A digital resolution target was generated with 15-pixel-wide bars and varying gap sizes (1, 3, 5, 7, 9 pixels). Ghost-GPT successfully resolved features with a 5-pixel gap. In our experimental setup, the Air Force resolution target's 23-pixel feature between a pair of stripes corresponds to a physical size of 0.355 mm, indicating that Ghost-GPT achieves a resolution of approximately 0.077 mm. At

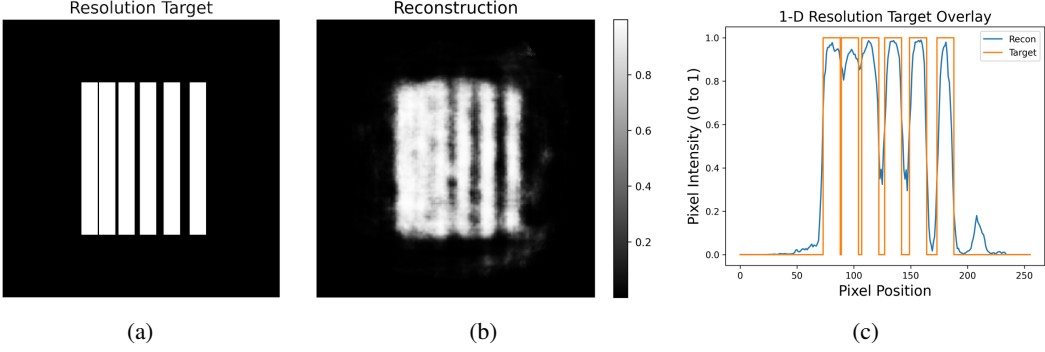

Figure 10: Simulated resolution testing of Ghost-GPT. (a) True resolution target with increasing pixel gaps (b) Ghost-GPT reconstruction results (c) Line profile comparing the ground truth and model output, obtained by scanning along a horizontal cross-section midway through the images.

this resolution, our ghost imaging model is well-suited for visualizing a broad range of biological structures, including tissue architecture such as blood vessels, skin layers, and muscle fibers. It also enables observation of developmental forms in small organisms like embryos and larvae, as well as structural features in plant tissues such as roots and leaves.

## E  Code Availability

The scripts, specific training/validation dataset, experimental dual-comb speckle patterns, and target measurements are available at: Code Repository Link. Instructions for running the scripts can be found in the `README.txt` file, and the required libraries and dependencies are specified in the `environment.yml` file.

## F  Broader Impacts

Although there are no evident potential negative societal impacts associated with dual-comb, transformer based ghost imaging, there are many notable positive impacts. These include advances in the health industry, specifically biomedical image quality, speed, and adaptability as clearer images can enable earlier disease detection and more precise surgical guidance. On top of better medical image quality, the integration of deep learning into reconstruction imaging heavily benefits endoscopy devices as it could allow for less invasive fiber-based procedures. In particular, transformer based ghost imaging can also boost overall accuracy by learning and compensating for complex patterns with environmental noise, producing high quality images even in low light or highly scattering conditions.

## G  Limitations

While Ghost-GPT demonstrates high reconstruction fidelity, computational efficiency, and robustness to noise, there are several limitations with our current work. First, the model performance is contingent on the quality and representativeness of the calibration speckle patterns; any deviations in experimental conditions may lead to distribution shift, potentially reducing reconstruction accuracy. Second, although the model generalizes well to synthetic and experimental data within the resolution constraints of our setup, it has not been benchmarked on more complex, naturalistic scenes or dynamic objects beyond the training domain. Additionally, while the use of synthetic training data from MNIST and Omniglot enables rapid prototyping, the domain gap between these datasets and real biological or clinical targets may necessitate fine-tuning with domain-specific data. Third, our current framework is only comparing our model against classical algorithms.

In this work, we focused our comparison on classical ghost imaging reconstruction algorithms such as Differential Ghost Imaging, the Moore-Penrose pseudoinverse, and FISTA. However, further benchmarking against advanced generative or super-resolution models—such as diffusion models,

adversarial networks, or U-Nets —could offer insight into alternative priors and reconstruction strategies (i.e. more complex loss functions) that may enhance fidelity or reduce artifacts. We emphasize that this work represents an early proof-of-principle for dual-comb ghost imaging using a transformer-based architecture and should be interpreted as a foundation for future improvements. Continued development will include more sophisticated training data, model architectures, and experimental protocols to fully exploit the potential of this novel imaging paradigm.

