# OpenReview forum: "Dual-Comb Ghost Imaging with Transformer-Based Reconstruction for Optical Fiber Endomicroscopy"
_NeurIPS.cc/2025/Conference — NeurIPS 2025 poster_

### Official Review · Reviewer_TRQN · 2025-06-25

**Clarity:** 3
**Significance:** 3
**Originality:** 3
**Rating:** 4
**Confidence:** 3

**Summary:**

This paper presents a novel approach to endoscopic imaging by combining dual-comb interferometry, ghost imaging, and a transformer-based reconstruction model (Ghost-GPT). The key innovation lies in leveraging optical frequency combs to generate uncorrelated speckle patterns via wavelength-division multiplexing through a single-core fiber, enabling snapshot detection with a single-pixel detector. The proposed Ghost-GPT model achieves high-fidelity image reconstruction at ultra-low sampling ratios and outperforms classical algorithms in both speed and accuracy.

**Questions:**

1) Is the proposed method limited to static samples, or can it also handle dynamic imaging scenarios?
2) Are there any constraints on spatial resolution? Specifically, is cellular-level resolution achievable with this approach?
3) Could you clarify why comparisons with state-of-the-art generative models were not included?
4) Given that dual-comb hardware demands precise alignment, does this significantly hinder its feasibility for real-world applications?
5) Does the model need to be retrained for different types of data? If so, how long does the training process take?

**Ethical Concerns:**

["NO or VERY MINOR ethics concerns only"]

**Final Justification:**

The rebuttal clarified that the experiments were designed to validate the feasibility of the proposed method. This kind of interdisciplinary research is likely to attract interest from the NeurIPS community.

**Limitations:**

yes

**Quality:**

3

**Strengths And Weaknesses:**

Strengths:
1) Combining dual-comb interferometry with ghost imaging allows snapshot imaging without mechanical scanning, making the system smaller and cheaper.
2) Ghost-GPT is faster and produces better image quality than traditional methods.
3) The small fiber probe and computational imaging help solve common problems with standard endoscopes, making it useful for minimally invasive diagnosis.

Weaknesses：
1) Validation is limited to synthetic data (MNIST/Omniglot) and simple targets (USAF charts), with untested performance on complex biological tissues.
2) The reconstruction accuracy appears to rely heavily on pre-calibrated speckle patterns, which raises concerns about the system's robustness in the presence of fiber bending, mechanical perturbations, or environmental variations.

---

> ### Author Rebuttal · Authors · 2025-07-31
>
> Dear Reviewer TRQN,
>
> Thank you for your insightful review and feedback. In the following, we will address your comments and questions:
>
> **Weakness 1: Validation is limited to synthetic data (MNIST/Omniglot) and simple targets (USAF charts), with untested performance on complex biological tissues.**
>
> Thank you for the comment. While the validation is limited to synthetic data and simple targets, we emphasize that this work represents the first optical experimental demonstration of video-rate ghost imaging enabled by dual-comb interferometry. Due to experimental constraints—specifically the limited number of comb lines (~200) and the speckle correlation width—we tested Ghost-GPT primarily on simple binary datasets. This limited resolution is not a fundamental limitation of the approach. We are actively working to increase the number of comb lines to 1,000–10,000, which will allow evaluation on more realistic grayscale datasets and eventually complex biological tissues.
>
> **Weakness 2: The reconstruction accuracy appears to rely heavily on pre-calibrated speckle patterns, which raises concerns about the system's robustness in the presence of fiber bending, mechanical perturbations, or environmental variations.**
>
> In our current optical experiment, speckle patterns are generated using a hybrid optical fiber consisting of a single-mode fiber (SMF) spliced to a multi-mode fiber (MMF). Bending the SMF does not affect the speckle patterns; however, bending the MMF does affect the speckle pattern and requires recalibration. To improve the robustness of the generated speckle patterns, we are exploring an all-single-mode fiber approach by coating the SMF tip with either a rough multi-mode scattering layer or multi-layered metasurfaces to create a compact wavelength-dependent speckle/structured light generator. We expect this method to minimize the effects of environmental variations and improve the stability and reliability of the generated speckle patterns. Please refer to Shin, J et al., "Single-pixel imaging using compressed sensing and wavelength-dependent scattering," Opt. Lett. 41, 886-889 (2016), who used a scattering layer of TiO2  to achieve wavelength dependent scattering on a SMF with a 0.1 nm spectral step size.
>
> **Question 1: Is the proposed method limited to static samples, or can it also handle dynamic imaging scenarios?**
>
> As aforementioned, the greatest advantage of our dual comb interferometry setup is that we are able to achieve video-rate capable imaging. We verified this in our most recent work by moving a USAF target at a rate of 2.4 mm/s and recorded a 60 FPS movie of the target moving using our experimental optics setup. In addition, our most current results show imaging frame rate up to 1000 Hz.
>
> **Question 2: Are there any constraints on spatial resolution? Specifically, is cellular-level resolution achievable with this approach?**
>
> Spatial resolution is primarily determined by the speckle grain size; smaller grains yield higher resolution. By adjusting the fiber position, we can dynamically control grain size and thus resolution. Previous work by Zhu et al. demonstrated cellular-level resolution in ghost imaging via a strong magnifying objective lens (20x - 40x) in conjunction with a digital micromirror device (DMD). Similarly, our setup can achieve this by placing a strong objective lens at the fiber tip—though this reduces modularity.
>
> To maintain modularity while achieving cellular resolution, we propose integrating a metalens—a flat, nanostructured ultra-thin lens—at the fiber tip to focus light using subwavelength structures. Previous works in metamaterials by Khorasaninejad et al, show that metalens are capable of achieving magnifications up to 170x, highlighting their potential to enable compact, cellular-resolution imaging at the tip of an optical fiber.
>
> References :
>
> Zhu, X.-H., et al. “Live cell imaging and classification via microscopic ghost imaging.” Phys. Rev. Applied, 23(5), 2025.
>
> Khorasaninejad, M et al., Metalenses at visible wavelengths: Diffraction-limited focusing and subwavelength resolution imaging. Science, 352, 1190-1194(2016)
>
> **Question 3: Could you clarify why comparisons with state-of-the-art generative models were not included?**
>
> Regarding the lack of comparison with other advanced algorithms, we would like to clarify that our primary goal was to demonstrate that the Ghost-GPT model enables fast, high-fidelity imaging when integrated with our novel experimental approach. Additionally, the limited number of comb lines in our current experimental optics setup constrained the dataset complexity, making it less suitable for comparisons with other advanced generative models using complex grayscale image datasets.
>
> However, we agree that such comparisons would be valuable. Since submission, we have conducted these comparisons and found that our transformer-based Ghost-GPT model outperforms U-Net, CNN, and diffusion models of similar size in terms of training speed and reconstructed image fidelity. Specifically, our transformer model outperforms CNN and U-Net variants of similar size by approximately 10% in mean squared error on the validation set, and surpasses a Diffusion-based variant by over 50%. We will include these results in the camera-ready manuscript.
>
> **Question 4: Given that dual-comb hardware demands precise alignment, does this significantly hinder its feasibility for real-world applications?**
>
> The dual-comb is generated from a robust fiber-connectorized system and delivered to the optical fiber probe via single-mode fiber, requiring no precise alignment. The level of alignment needed is not significantly different from that of conventional ghost imaging. While alignment between the single-core optical fiber probe and the single-pixel detector remains important in the current transmissive configuration, a reflective configuration—integrating the single-core fiber with the single-pixel detector and a beam collimator to detect the reflected beam from the target object—could make beam collection much more efficient without precise alignment. With such a reflective-mode optical fiber probe, our dual-comb compressive imaging approach is feasible for real-world applications.
>
>
> **Question 5: Does the model need to be retrained for different types of data? If so, how long does the training process take?**
>
> Our model trains in approximately 1 hour and 30 minutes for 50 epochs on a single NVIDIA A6000 GPU. However, as shown in Figure 7b, performance saturates after about 10 epochs, meaning training time can be reduced to ~15 minutes without significant loss in reconstruction quality. When adapting to new data — particularly different speckle patterns — the model does not require full retraining. As shown in Appendix D, Figure 9c, initializing with pretrained weights allows the model to converge on a new speckle configuration in just 1–2 epochs, requiring only a few minutes. This demonstrates that the method is both computationally efficient and highly adaptable to different measurement conditions.
>
>
> Thank you again for your thoughtful review. We look forward to improving our manuscript with your valuable suggestions.

---

> > ### Comment · Reviewer_TRQN · 2025-08-05
> >
> > I thank the authors for addressing my comments, and I would like to keep my original score.

---

### Official Review · Reviewer_QCan · 2025-06-29

**Clarity:** 3
**Significance:** 2
**Originality:** 2
**Rating:** 4
**Confidence:** 4

**Summary:**

This work introduces a method for endoscopic imaging using ghost imaging. Building on a large corpus on existing work on ghost imaging, the key contribution of this work is a transformer-based reconstruction method that allows them to sample at relatively low rates. This makes it possible for the authors to design sensing matrices that allow for moderate resolution imaging through a single fiber. The method is compared to conventional compressed sensing methods that have been proposed in the earlier phases of this research direction.

**Questions:**

- How does the proposed transformer compare to existing transformers proposed for ghost imaging? The design seems to follow vanilla transformers. Please clarify.

- What is the runtime/efficiency of the method?

- Which application warrants the evaluation on the low-resolution images discussed in the paper?

**Ethical Concerns:**

["NO or VERY MINOR ethics concerns only"]

**Final Justification:**

The authors provided additional experimental validation. As such, I have increased my rating. While the margins are still relatively incremental, the additional experiments are promising.

**Limitations:**

Yes

**Quality:**

2

**Strengths And Weaknesses:**

Strengths:
- The paper is well-written and provides a comprehensive overview of ghost imaging and speckle-based compressive imaging.
- The optical setup and experiments are well executed and, at low resolution, convincingly demonstrate that the method works as advertised.

Weaknesses:
- The reconstruction network is not evaluated in detail and ablation experiments are missing in the manuscript.
- The novelty of the method is limited as there is a large body of work that focuses on transformer/GPT-like reconstruction methods for Ghost imaging, some are listed below:

Ren, Wenhan, et al. "Ghost translation: an end-to-end ghost imaging approach based on the transformer network." Optics Express 30.26 (2022): 47921-47932.

Liang, Jiayuan, Yu Cheng, and Jiafeng He. "Transformer-based flexible sampling ratio compressed ghost imaging." Engineering Analysis with Boundary Elements 170 (2025): 106050.

Chen, Y., An, H., Sun, Z., Tian, T., Chen, M., Spielmann, C., & Li, X. (2025). Large model enhanced computational ghost imaging. arXiv preprint arXiv:2503.08710.

He, Y., Zhou, Y., Yuan, Y., Chen, H., Zheng, H., Liu, J., ... & Xu, Z. (2022). TransUNet-based inversion method for ghost imaging. Journal of the Optical Society of America B, 39(11), 3100-3107.

Manni, Mathieu, Dmitry Karpov, K. Joost Batenburg, Sharon Shwartz, and Nicola Viganò. "Noise2Ghost: Self-supervised deep convolutional reconstruction for ghost imaging." arXiv preprint arXiv:2504.10288 (2025).

Wang, Mingcong, Linqing Jang, Hexiao Wang, and Minghui Ma. "Computational Ghost Imaging via Speckle Illumination and Deep Learning Under Low Sampling Rates." In 2024 5th International Symposium on Computer Engineering and Intelligent Communications (ISCEIC), pp. 567-572. IEEE, 2024.

None of these existing methods are evaluated and compared to. The evaluation is focused on outdated methods that unfortunately offer only strawman comparisons for the method.

- Esoteric application: The application demonstrated in this work is rather esoteric (see related work above) and offers little insight into the scientific findings that help design better reconstruction algorithms, too. With the low resolution results demonstrated in this work, the method is also not practical.

---

> ### Author Rebuttal · Authors · 2025-07-31
>
> Dear Reviewer Qcan,
>
> Thank you for your insightful review and feedback. In this rebuttal, we would like to clarify the novelty of our work and explain the constraints imposed by the experimental setup. Below, we address your comments and questions:
>
> **Weakness 1: The reconstruction network is not evaluated in detail and ablation experiments are missing in the manuscript.**
>
> We appreciate the reviewer’s suggestion regarding ablation studies. Our primary objective in this work was to demonstrate that the Ghost-GPT model enables fast, high-fidelity imaging when combined with our novel experimental approach. Due to the limited number of comblines in our initial setup, the dataset complexity was constrained, making it less suitable for a broad range of ablation studies. As a result, we focused on analyzing the most critical components of our model architecture—namely, the number of attention heads and the size of the embedded speckle patterns. This analysis is presented in Appendix C (Figure 7), where we show that increasing both of these factors leads to improved model performance.
>
> We agree that a deeper study of our architecture would be valuable, such as comparisons with other state-of-the-art models (Diffusion, CNN, U-Net), examining the effects of removing speckle pattern embeddings, or experimenting with alternative loss metrics. In our ongoing work, we address these points. Initial results show that our transformer model outperforms CNN and U-Net variants of similar size by approximately 10% in mean squared error on the validation set, and surpasses a Diffusion-based variant by over 50%. Additionally, removing the speckle pattern embedding led to unstable training and failure to converge. We have also explored alternative loss functions including total variation (TV) loss and Multi-Scale Structural Similarity Index Measure (MS-SSIM). It is also observed that the regularised loss function with the optimised weights for each term gives a much better reconstruction. We will incorporate these experiments in the updated manuscript.
>
> **Weakness 2: The novelty of the method is limited as there is a large body of work that focuses on transformer/GPT-like reconstruction methods for Ghost imaging, some are listed below:**
>
> Thank you for this important comment. While prior studies have applied transformer architectures to ghost imaging, our work introduces two key novelties. First, the use of dual-comb interferometry in optical fiber ghost imaging enables fast, video-rate imaging even with zero-dimensional optical front-end hardware (e.g., single-core fiber and a single-pixel detector). Without dual-comb interferometry, achieving video-rate (greater than 30 FPS) ghost imaging would be extremely challenging, if not impossible. Second, our Ghost-GPT algorithm achieves high-fidelity, rapid image reconstruction from a limited number of fiber-generated speckle patterns.
>
> Many existing works, including the six papers listed by the reviewer, assume or generate a large number of speckle patterns or structured light patterns via spatial light modulators (SLMs) or digital micromirror devices (DMDs). SLMs and DMDs produce relatively simple binary light patterns, which are easier to model and reconstruct compared to the complex and chaotic speckles produced by our optical fiber system. However, the bulkiness and slow switching speeds of SLMs and DMDs render them impractical for endoscopic applications.
>
> Compared to SLMs and DMDs, generating a large number of uncorrelated speckle patterns from single-core optical fibers is not technically straightforward. As the first experimental demonstration of dual-comb interferometry in ghost imaging, our setup is currently limited to about 200 speckle patterns due to experimental constraints (e.g., the FWHM of speckle correlation and the number of comb lines). In comparison, related works (e.g., Chen et al., Liang et al.) use a much larger number of speckle patterns and sampling ratios 4 to 35 times greater, with simpler speckle patterns. Therefore, direct reconstruction quality comparisons without accounting for these experimental limitations are not entirely fair. Moreover, given the significant novelty of dual-comb interferometry in ghost imaging and its potential for further experimental advances, it is difficult to directly compare our work with previous SLM- or DMD-based studies combined with transformer- or GPT-like reconstruction methods solely on the algorithmic aspect.
>
> **Weakness 3: Esoteric application: The application demonstrated in this work is rather esoteric (see related work above) and offers little insight into the scientific findings that help design better reconstruction algorithms, too. With the low resolution results demonstrated in this work, the method is also not practical.**
>
> We thank the reviewer for the feedback. We address the esoteric nature of the work in the previous comment, so we will address the low resolution results demonstrated in this work. Again, we emphasize that this is the first experimental demonstration of video-rate capable ghost imaging with dual comb interferometry, and thus, is the primary contribution of this work. While we agree that the reported resolution in the manuscript is insufficient for many practical applications, it represents an important proof-of-concept demonstration of dual comb based ghost imaging and demonstrates its potential. Furthermore, while the limited resolution is a result of the limited number of combines, it is not a fundamental limitation. We are currently working on increasing the number to 1,000~ 10,000, which will allow testing with more realistic and practical grayscale datasets.
>
> **Question 1: How does the proposed transformer compare to existing transformers proposed for ghost imaging? The design seems to follow vanilla transformers. Please clarify.**
>
> We thank the reviewer for this important question. Although our model is based on the standard transformer architecture, we introduce task-specific adaptations to suit the unique structure of fiber based ghost imaging data. In particular, we implement a physics-aware embedding strategy that maps compressed measurements into a structured token space integrating sampling mask information and coarse positional priors, enabling more effective reconstruction in this domain.
>
> **Question 2: What is the runtime/efficiency of the method?**
>
> We appreciate the reviewer’s interest in runtime performance. On a single NVIDIA A6000 GPU, the model reconstructs a 256×256 image in approximately 8 ms per frame.
>
> **Question 3: Which application warrants the evaluation on the low-resolution images discussed in the paper?**
>
> We appreciate the reviewer’s question regarding the use of binary, low-resolution images. As the first experimental demonstration of dual-comb ghost imaging, we are limited to 200 comblines which constrains the achievable resolution and dynamic range. To emphasize, this resolution constraint is not a fundamental limitation of our approach. We are actively working in the lab to increase the number of comb lines to the range of 1,000–10,000, which will enable evaluation on more realistic grayscale datasets and, ultimately, complex biological tissue samples.
>
> Despite this limitation, such the current spatial resolution (on the order of millimeters) is still relevant to several practical applications. In needle-based confocal laser endomicroscopy (nCLE), probe miniaturization often results in lower-resolution images sufficient for binary detection of lesions or contrast uptake. Outside medicine, low-resolution binary imaging is widely used in industrial inspection, such as detecting missing components on printed circuit boards or contaminants on conveyor belts, where presence/absence classification is sufficient. Additionally, in embedded or radiation-constrained environments—like pipeline inspection or wearable sensors—binary, low-resolution imaging reduces data throughput enabling real-time and robust operation. Therefore, although constrained by hardware, the imaging scale remains relevant for multiple important real-world applications.
>
>
> Thank you again for your constructive comments. We look forward to incorporating your suggestions to improve our manuscript.

---

> > ### Comment · Reviewer_QCan · 2025-08-05
> > **Ablation Experiments**
> >
> > Thank you for the rebuttal. Can you provide more details on the ablation experiments you performed, especially regarding MSE. The improvement of 10% seems rather incremental so it would be helpful to understand the experiment in detail here to see what exactly on which dataset is measured. Also, how do these experiments scale with the model capacity of the baseline models (U-Net)?

---

> ### Author Response · Authors · 2025-08-08
>
> Thank you for the follow-up and for the comment about incremental performance. The 10% MSE improvement reported in the rebuttal was a conservative estimate (or lower bound) based on initial comparisons over a small subset of validation images, and it underrepresents the actual performance gains of our model. When evaluated over the full validation dataset, Ghost-GPT outperforms CNN and U-Net by 30.5% and 19.3%, respectively, in terms of relative MSE — all under identical training settings, including number of epochs. We believe this modest improvement is largely due to the simplicity and sparsity of the binary-valued images (MNIST and OMNIglot), which contain large background regions with minimal structure. In such settings, even conventional models like U-Net perform near their optimal capacity, limiting the headroom for further improvement.
>
> While we have not conducted explicit scaling experiments for U-Net, we did evaluate Ghost-GPT at both 20 million and 264 million parameters. On the same dataset, the larger model achieved an MSE of 0.007 compared to 0.00742 for the smaller variant — a 5.66% improvement. This small gain supports the view that model scaling yields diminishing returns on this dataset. Based on these observations, we expect U-Net to follow a similar scaling trend, with limited changes in performance as we either increase or decrease the number of parameters.
>
> Lastly, if you still have questions regarding the novelty or contribution of our work, we kindly refer you to our additional responses to the other reviewers. Thank you very much for the invaluable comments and fruitful discussion.

---

### Official Review · Reviewer_T9rk · 2025-07-03

**Clarity:** 2
**Significance:** 2
**Originality:** 2
**Rating:** 3
**Confidence:** 4

**Summary:**

This paper proposes a transformer-based compressive-sensing reconstruction algorithm for optical fiber endomicroscopy. Experiments have demonstrated the effectiveness and efficiency of the proposed algorithm.

**Questions:**

1) What are the characteristics of the observation matrix ${\Psi}$ in this problem. The authors should give a brief discussion on this, because the observation matrix strongly influences the quality of reconstruction.
2) In Table I, the authors compare the computational time of different algorithms but do not specify the platform that these algorithms run on. Do these algorithms run on a CPU, GPU? or some of them run on a CPU and some of them run on a GPU?
3) Experiments are conducted on simulated data. The performence of the proposed algorithm on real data is not clear.

**Ethical Concerns:**

["NO or VERY MINOR ethics concerns only"]

**Final Justification:**

Thank you for the effort in considering and replying to all the points mentioned in the review. The rebuttal has improved my understanding of the contributions of the paper, but limitations on novelty and applicability of the proposed method still exist. Therefore I would like to keep my original score.

**Limitations:**

1) The novelty is fair. What is the biggest difference between the proposed transformer-based algorithm with existing compressive-sensing reconstruction algorithms? Why should the proposed algorithm be more effective than existing algorithms?
2) Experimental results are not sufficiently convincing. The compared algorithm Moore-Penrose Psuedo Inverse should not be counted as a compressive-sensing reconstruction algorithm. More advanced algorithms should be compared in experiments.
3) The effectiveness of the proposed algorithm on real data has not been demonstrated.

**Paper Formatting Concerns:**

no major formatting issue.

**Quality:**

3

**Strengths And Weaknesses:**

Strengths:
Optical fiber endomicroscopy is an interesting problem that very suits to apply compressive-sensing algorithms, the authors demonstrate the possibility to improve the imaging quality of optical fiber endomicroscopy by employing transformer-based deep models.

Weakness:
1) The novelty is fair. What is the biggest difference between the proposed transformer-based algorithm with existing compressive-sensing reconstruction algorithms? Why should the proposed algorithm be more effective than existing algorithms?
2) Experimental results are not sufficiently convincing. The compared algorithm Moore-Penrose Psuedo Inverse should not be counted as a compressive-sensing reconstruction algorithm. More advanced algorithms should be compared in experiments.
3) It would be better if the authors could further clarify the difference between optical fiber endomicroscopy between classical compressive-sensing problems, i.e., in aspects of image prior, encoding matrix, input measurements, and so on.

---

> ### Author Rebuttal · Authors · 2025-07-31
>
> We thank the reviewer for the careful reading and constructive feedback. In this rebuttal, we would like to clarify the novelty of our work and explain the constraints imposed by the experimental setup.
>
> **Response to the reviewer’s comments on weaknesses and limitations:**
>
> **Weakness & Limitation 1: The novelty is fair. What is the biggest difference between the proposed transformer-based algorithm with existing compressive-sensing reconstruction algorithms? Why should the proposed algorithm be more effective than existing algorithms?**
>
> The novelty of our work is twofold: (i) adopting the dual-comb interferometry method for the first time in a ghost imaging experiment and (ii) combining it with a transformer-based model that enables fast, high-fidelity imaging. First, the use of dual-comb interferometry in ghost imaging allows rapid, video-rate optical detection. Second, the Ghost-GPT algorithm supports high-fidelity, fast image reconstruction. Regarding the novelty of our Ghost-GPT model, although it is now well known in the community that deep learning models often outperform iterative reconstruction algorithms in inverse problems, our Ghost-GPT model uses both speckle patterns and bucket sums as input tokens, enhancing training speed and convergence while achieving higher image fidelity than non-ML algorithms. We also note that the reviewer’s evaluation appears to focus mostly on the ML-algorithm side, overlooking both the dual-comb optical experiment aspect and the experimental constraints that influenced the choices for ML model design and datasets.
>
> To be specific, as this is the first experimental demonstration of dual-comb interferometry in ghost imaging, we were limited to approximately 200 speckle patterns due to experimental constraints (e.g., the FWHM of speckle correlation and the number of comb lines). This restriction limited us to testing Ghost-GPT on simple, binary datasets. Importantly, the small number of speckle patterns is not a fundamental limitation. Rather, the slower pace of experimental improvements, compared to computational algorithms, has constrained our current results. As a follow-up experiment, we are actively working on increasing the number of uncorrelated speckle patterns to the 1,000-10,000 range by reducing the FWHM of speckle correlation and spectral broadening of the optical frequency combs. This will enable testing with more realistic grayscale datasets, as shown in Chen, Yifan, et al. "Large model enhanced computational ghost imaging." arXiv preprint arXiv:2503.08710 (2025). We hope this clarifies why we chose simple binary image datasets rather than more complex image datasets to demonstrate the feasibility of our novel experimental method, and we believe it also addresses the reviewer’s concerns about the effectiveness of the proposed algorithm on real data.
>
> **Weakness & Limitation 2: Experimental results are not sufficiently convincing. The compared algorithm Moore-Penrose Psuedo Inverse should not be counted as a compressive-sensing reconstruction algorithm. More advanced algorithms should be compared in experiments.**
>
> Regarding the concerns about the lack of comparison with other advanced algorithms, we would first like to clarify that our primary goal was to demonstrate that the Ghost-GPT model enables fast, high-fidelity imaging when combined with our novel experimental method using dual-comb interferometry, rather than to claim superiority over all DL models. Nevertheless, since the submission, we have compared our model with CNN, U-Net, ViT (on which our Ghost-GPT model is based), and other DL models using the same simple binary datasets. These initial results show that transformer-based models generally perform better in terms of training speed and convergence. However, a more rigorous comparison will require testing the models with a larger number of speckle patterns and more complex datasets, as discussed above.
>
> **Weakness 3: It would be better if the authors could further clarify the difference between optical fiber endomicroscopy between classical compressive-sensing problems, i.e., in aspects of image prior, encoding matrix, input measurements, and so on.**
>
> Compared to previous compressive imaging demonstrations that use spatial light modulators or digital micromirror devices to generate a large number of imaging basis patterns, our dual-comb compressive imaging approach for optical fiber endomicroscopy uses a limited number of speckle patterns due to experimental constraints (though this is not a fundamental limitation) and minimally invasive optical front-end hardware (e.g., single-core fiber and a single-pixel detector). To enable video-rate imaging with such hardware-size constraints, we adopted dual-comb interferometry for fast snapshot detection of bucket-sum signals. Additionally, because of the limited number of available speckle patterns in the current experiment, our algorithmic analysis was restricted to simple binary image datasets.
>
> **Limitation 3: The effectiveness of the proposed algorithm on real data has not been demonstrated.**
>
> The model is trained on simulated bucket-sum measurements using known image datasets and the exact speckle patterns used experimentally, and is then applied to real bucket-sum data acquired through dual-comb measurements to reconstruct the test target image. This real-world optical experiment is conducted to evaluate the feasibility of dual-comb interferometry in ghost imaging, not to test the Ghost-GPT algorithm itself. Due to the limited number of speckle patterns, we used simple binary datasets for model training and simple binary patterns (e.g., the USAF1951 resolution target) as real-world target objects in the current work. However, as discussed earlier, with an increased number of speckle patterns, we will be able to use more complex grayscale image datasets for model training, and we expect to image real biological samples.
>
> **Answers to the reviewer’s questions:**
>
> **Question 1:  What are the characteristics of the observation matrix in this problem. The authors should give a brief discussion on this, because the observation matrix strongly influences the quality of reconstruction.**
>
> In our compressive imaging work, the set of uncorrelated speckle patterns constitutes the observation matrix. This observation matrix is random and incoherent, with non-negative real-valued elements. The correlations between the speckle patterns are shown in Figure 2(b). The number of uncorrelated speckle patterns is limited by the number of comb lines and the speckle correlation bandwidth, and this finite number of uncorrelated patterns imposes a limit on the achievable resolution and reconstruction quality.
>
> **Question 2: In Table I, the authors compare the computational time of different algorithms but do not specify the platform that these algorithms run on. Do these algorithms run on a CPU, GPU? or some of them run on a CPU and some of them run on a GPU?**
>
> The algorithms were run on a Linux workstation using an AMD Ryzen Threadripper 3990X 64-Core Processor and a single NVIDIA A6000 GPU. To be more specific, the differential ghost imaging (DGI) as well as our Ghost-GPT model ran on GPU. The pseudoinverse (PI) and fast iterative shrinkage-thresholding algorithm (FISTA) ran on CPU. We will make these distinctions clear in the revised manuscript.
>
> We will also include updated tests on GPU enabled versions of these algorithms for a fairer comparison. Initial tests on a PyTorch-GPU version of the PI show a computational time of 157 ms +/- 3.27 ms, while a GPU version of FISTA through the pylops_gpu library show worse performance at an average computational speed of 10.6 seconds +/- 60.2 ms. We attribute this discrepancy in FISTA primarily to data transfer overheads between CPU and GPU in the pylops_gpu implementation.
>
> Despite these updated benchmarks, our original claim remains valid: Ghost-GPT model provides the fastest reconstruction speed among the compared approaches.
>
> **Question 3: Experiments are conducted on simulated data. The performence of the proposed algorithm on real data is not clear.**
>
> The Ghost-GPT model was trained using simulated data and tested on both simulated data (Figure 4) and experimental optical measurement data (Figure 5). Because the number of speckle patterns was limited to approximately 200, we used the USAF1951 resolution target pattern for the optical experiment, as it is simple enough to reconstruct with a limited number of imaging basis patterns. As shown in the 2025 arXiv paper, we expect that grayscale imaging will be possible with an increased number of speckle patterns (1,000–10,000) and more realistic, complex datasets. Increasing the number of uncorrelated speckle patterns in the dual-comb experiment is an ongoing effort in our lab.

---

> > ### Comment · Reviewer_T9rk · 2025-08-08
> >
> > Thank you for the effort in considering and replying to all the points mentioned in the review. The rebuttal has improved my understanding of the contributions of the paper, but limitations on novelty and applicability of the proposed method still exist. Therefore I would like to keep my original score.

---

> ### Comment · Reviewer_T9rk · 2025-08-08
>
> Thank you for the effort in considering and replying to all the points mentioned in the review. The rebuttal has improved my understanding of the contributions of the paper, but limitations on novelty and applicability of the proposed method still exist. Therefore I would like to keep my original score.

---

> > ### Author Response · Authors · 2025-08-09
> > **Further clarification on the comment "limitations on novelty and applicability of the proposed method still exist"**
> >
> > Thank you for carefully reading our rebuttal. Regarding your comment, “limitations on novelty and applicability of the proposed method still exist”, we would like to clarify this point again.
> >
> > Prior end-to-end transformer works in ghost imaging (for example, Ren, Wenhan, et al. "Ghost translation: an end-to-end ghost imaging approach based on the transformer network." Optics Express 30.26 (2022): 47921-47932.) were trained using solely bucket sums measurement, without the embedding of the per-measurement speckle patterns. In contrast, our Ghost-GPT model follows the end-to-end approach but introduces a key departure from prior work: we encode paired tokens that jointly contain the speckle pattern and its corresponding bucket sum. This allows the transformer to attend over both spatial structure (from the pattern) and measured intensity (from the bucket) in a unified representation. This speckle-aware model leverages more information during training, enabling better convergence and faster learning. This means the improved stability of training as well as the smaller number of epochs necessary for the model’s loss to asymptote. Training runs without speckle pattern embedding were less stable — they required more careful hyperparameter tuning and multiple attempts to obtain convergence in training— whereas training with the paired tokens converged more consistently in our setup. In addition, we observed the following in our experiments: the speckle patterns embedded into the tokens improves reconstruction fidelity (Fig. 7); specifically, as we increase the size of the speckle pattern embedding, the MSE decreases.
> >
> > In addition, our novelty lies in the integrated combination of dual-comb detection, optical fiber-based imaging, and advanced ML models. Simply put, our work exemplifies the co-design of imaging hardware and methodology (dual-comb ghost imaging) with advanced ML models, with the goal of achieving video-rate endoscopic imaging under extreme hardware constraints. The ultrafast detection enabled by the dual-comb source, together with the miniature imaging conduit provided by the optical fiber, addresses key bottlenecks that have restricted prior, and most importantly, useful implementations of ghost imaging.
> >
> > Most prior ML-only algorithmic works on ghost imaging assume that a large number of speckle patterns are measured simultaneously; however, this assumption is invalid because speckle data are typically acquired serially at a slow rate, making it impossible to image rapidly moving objects. From an ML contribution perspective, with the importance of high-quality data in ML research continuing to grow, our work validates the assumption of a simultaneous imaging matrix (speckle pattern) and enables high-quality, simultaneous data acquisition even for rapidly moving objects in real-world settings. We would like to emphasize that good imaging requires both real-time capability and high fidelity, and our work focuses on achieving both—not just the high-fidelity aspect—through the co-design of the entire imaging system (hardware + software).
> >
> > These advances open the door to practical applications of ML-enhanced video-rate ghost imaging—such as endomicroscopy—and thereby enable new opportunities for machine learning methods tailored to this emerging domain and expand the compressed sensing field as whole. By removing longstanding experimental bottlenecks, our hardware advances in ghost imaging pave the way for applying more advanced, domain-specific, and novel ML methods. Importantly, since our paper was submitted to the “Applications (e.g., vision, language, speech and audio, Creative AI)” track, we believe that application-specific or application-aware co-design of hardware and software is valuable, as opposed to software-only ML contributions.

---

### Official Review · Reviewer_wkCt · 2025-07-03

**Clarity:** 4
**Significance:** 3
**Originality:** 4
**Rating:** 5
**Confidence:** 5

**Summary:**

the paper proposes a supervised learning approach using a transformer based model for an inverse imaging problem. Experiments show that the proposed model demonstrates higher accuracy and speed on synthetic datasets and some select real world samples compared to iterative reconstruction algorithms.

**Questions:**

1. **Choice of training data**: The authors are requested to clarify the motivation behind using MNIST/Omniglot for training. Including results on more realistic or diverse datasets would strengthen the paper considerably.

2. **Real-world data**: For the experiments on real-world data (lines 222–226), to better appreciate the significance of these experiments, it would be helpful to discuss the importance of this real world data. A discussion is also needed around what dataset should be used to train the model for the real data? If the model is trained on synthetic digits, how would it perform on real data? This clarification is important for assessing the generalization capabilities of the proposed method.

3. **Comparative baselines**: It would be helpful to include comparisons with other deep learning-based reconstruction approaches, including CNNs or transformers of varying capacities. This would allow for a more comprehensive evaluation of the proposed model's effectiveness.

4. **Reconstruction speed and system-level constraints**: Lines 216–221 discuss the speed of ~8 ms, but it would be useful to understand how this compares to the theoretical speed limit set by the system’s repetition rate. Additionally, the statement about the Nyquist condition and sampling/frame rate trade-off could benefit from further explanation tailored to a general ML audience.

5. **Noise robustness claims**: The robustness analysis appears to be based on a single example. To make a strong claim about the robustness of the proposed method, experiments need to be performed on a variety of samples.

**Ethical Concerns:**

["NO or VERY MINOR ethics concerns only"]

**Final Justification:**

- Rebuttal clarified dataset acquisition (optical, not simulated), justified dataset choices and elaborated on the novelty of the work.
- Rebuttal clarified how the existing experiments are sufficient and more validation is beyond the scope of this work,
- The broader themes and novel applications shown in this work would be interesting for the broader Neurips community. And most of the required modifications to the presentation of the paper (as discussed in rebuttal phase), are possible in the final revision of the paper.
- Such interdisciplinary works should receive more encouragement in the applications track of Neurips

**Limitations:**

No. Please see above sections

**Quality:**

3

**Strengths And Weaknesses:**

**Strengths**

- The paper is clearly written and easy to follow, which aids accessibility for a broad ML audience.
- Technical concepts such as ghost imaging and limitations of existing endoscopic imaging techniques are introduced and explained in a way that is informative and approachable.
- The work introduces a novel and timely problem to the ML community, bridging an interesting gap between optics and machine learning.
- Experiments include a preliminary study on robustness to noise, which is an important consideration for practical applications.

**Weaknesses**

- **Limited originality in ML contribution**: While the application area is novel, the machine learning component—supervised learning using a transformer—follows a well-established pattern. It is now common knowledge in the community that deep learning models tend to outperform iterative reconstruction algorithms in inverse problems (e.g., compressive sensing, MRI), both in terms of accuracy and speed. As such, the methodological contribution on the ML side may be viewed as limited in novelty.

- **Use of overly simplistic datasets**: The training and evaluation are primarily conducted on MNIST and Omniglot, which are not shown to be representative of the patterns found in endoscopic images. These datasets have binary-valued objects and lack realistic shape complexity. In accordance with related literature, experiments on more complex datasets(such as CelebA or imagenet) would make the experiments more rigourous.

- **Limited diversity of baselines**: All baseline methods used for comparison are iterative algorithms. The lack of comparisons against more recent methods including other deep learning-based reconstruction methods (e.g., CNNs or other transformer variants) makes it difficult to assess how well the proposed model performs relative to recent advancements in the field.

- **Concerns about practical applicability**: It is unclear whether the model was trained on real-world data or only on MNIST/Omniglot. If the latter, the assumption that it would generalize to practical endoscopic images is questionable and should be justified. Without a sufficiently addressing this issue, the applicability of the method to real clinical or imaging scenarios remains uncertain.

---

> ### Author Rebuttal · Authors · 2025-07-31
>
> We thank the reviewer for the careful reading and constructive feedback. In this rebuttal, we would like to clarify the novelty of our work and explain the constraints imposed by the experimental optics setup.
>
> **Response to the reviewer’s comment on the weaknesses:**
>
> **Weakness 1: Limited originality in ML contribution**
>
> ​​We fully acknowledge that deep learning models are now widely appreciated for often outperforming iterative reconstruction algorithms in inverse problems. In our Ghost‑GPT approach, we combine speckle patterns and bucket sums as input tokens, which offers improved convergence and faster training. We believe this aspect brings a degree of novelty. To clarify further, we state that the main novelty of our work is twofold:
>  (i) adopting the dual-comb interferometry method for the first time in a ghost imaging experiment, and,
> (ii) combining it with a transformer-based model that enables fast, high-fidelity imaging.
> First, the use of dual-comb interferometry in ghost imaging allows rapid, video-rate optical detection. Second, the Ghost-GPT algorithm supports high-fidelity, fast image reconstruction.
>
> **Weakness 2: Use of overly simplistic datasets** &
> **Weakness 4: Concerns about practical applicability**
>
> To be specific, as this is the first experimental demonstration of dual-comb interferometry in ghost imaging, we were limited to approximately 200 speckle patterns due to experimental constraints (e.g., the FWHM of speckle correlation and the number of comb lines). This restriction limited us to testing Ghost-GPT on simple, binary datasets. Importantly, the small number of speckle patterns is not a fundamental limitation. Rather, the slower pace of experimental improvements, compared to computational algorithms, has constrained our current results. As a follow-up experiment, we are actively working on increasing the number of uncorrelated speckle patterns to the 1,000-10,000 range by reducing the FWHM of speckle correlation and spectral broadening of the optical frequency combs. This will enable testing with more realistic grayscale datasets, as shown in Chen, Yifan, et al. "Large model enhanced computational ghost imaging." arXiv preprint arXiv:2503.08710 (2025). We hope this clarifies why we chose simple binary image datasets rather than more complex image datasets to demonstrate the feasibility of our novel experimental method using dual-comb, and we believe it also addresses the reviewer’s concerns about practical applicability.
>
> **Weakness 3: Limited diversity of baselines**
>
> Regarding the concern about the limited diversity of baseline methods, our primary goal was to demonstrate that the Ghost-GPT model enables fast, high-fidelity imaging when combined with our novel experimental method, rather than to claim superiority over all DL models. Nevertheless, we agree such comparisons are valuable. Since submission, we have compared Ghost-GPT with state-of-the-art models (Diffusion, CNN, and U-Net) using the same binary datasets. Our transformer model outperforms CNN and U-Net variants of similar size by ~10% in mean squared error on the validation set and surpasses a diffusion-based variant by >50%. While our transformer-based models generally perform better in image fidelity and training speed, a more rigorous comparison will require testing with 1,000–10,000 speckle patterns and more complex datasets, as discussed above.
>
> As an aside, if we had focused solely on the image reconstruction algorithm—assuming a large number of orthogonal imaging basis patterns could be generated using spatial light modulators or digital micromirror devices (as is common in prior computational imaging works)—comparing Ghost-GPT with other DL algorithms would have been simpler. However, in the case of minimally invasive, video-rate optical fiber endomicroscopy using zero-dimensional optical front-end hardware (e.g., single-core fiber and single-pixel detector), achieving video-rate imaging without dual-comb interferometry would be extremely challenging, if not impossible, and generating a large number of uncorrelated speckle patterns is not technically straightforward. Given the significant novelty of dual-comb interferometry in ghost imaging and its potential for further experimental advances, we believe it is somewhat unfair to evaluate our work solely on the Ghost-GPT model’s novelty.
>
> **Responses to specific reviewer questions:**
>
> **Question 1: choice of training data** MNIST and Omniglot were selected because our current experimental system provides only ~200 independent speckle patterns; binary/sparse datasets avoid underdetermined grayscale reconstructions. This does not fundamentally limit the method. Increasing the number of uncorrelated speckle patterns (i.e., comb lines) is already in progress and will enable testing with more complex datasets. We will clarify this rationale.
>
> **Question 2: real-world data** The model is trained on simulated bucket-sum measurements using known image datasets and the exact speckle patterns used experimentally, and is then applied to real bucket-sum data acquired through dual-comb measurements to reconstruct the test target image. This real-world optical experiment is conducted to evaluate the feasibility of dual-comb interferometry in ghost imaging, not to test the Ghost-GPT algorithm itself. Due to the limited number of speckle patterns, we used simple binary datasets for model training and simple binary patterns (e.g., the USAF1951 resolution target) as real-world target objects in the current work. However, as discussed earlier, with an increased number of speckle patterns, we will be able to use more complex grayscale image datasets for model training, and we expect to image real biological samples.
>
> **Question 3: comparative baselines** As briefly mentioned earlier, we have compared Ghost-GPT with state-of-the-art models (Diffusion, CNN, and U-Net) using the same binary datasets since the manuscript submission. Initial comparison results show that our approach achieves superior performance in terms of image fidelity and training speed. Specifically, our transformer model outperforms CNN and U-Net variants of similar size by ~10% in mean squared error on the validation set and surpasses a diffusion-based variant by >50%. These results will be included in the final version.
>
> **Question 4: reconstruction speed and system-level constraints** The model’s rapid reconstruction speed is currently 8 ms per frame when run on Nvidia A6000 GPU, and it can be further improved with more powerful computational hardware. The fundamental speed limit of our dual-comb compressive imaging method is set by the repetition rate difference of the dual comb, which in our current experiments ranges from a few hundred Hz to several kHz. Since the manuscript submission, we have demonstrated imaging of moving objects at frame rates of 30 Hz–1 kHz using a 3 kHz repetition-rate difference. Therefore, the overall imaging speed of the current experiment is limited by the computational reconstruction, not by the optical sampling speed of the dual-comb method.
>
> Regarding the statement about the Nyquist condition and the sampling/frame rate trade-off, we will tailor the explanation for a general ML audience as follows:
>
> “Importantly, while the computational reconstruction speed can be further improved with better computing hardware, the fundamental limit of image reconstruction is set by the repetition rate difference $\Delta f_{rep}$ of the dual-comb, which is typically a few hundred Hz to several kHz in our experiments. With a larger $\Delta f_{rep}$, a much higher frame rate is possible. However, this requires high-bandwidth photodetection and introduces an inherent trade-off between the frame rate $f_{frame}$ and the sampling ratio $\beta$ (i.e., the number of comb lines $M$ divided by the image resolution $N^2$). According to the Nyquist condition in the context of the dual-comb method, the repetition-rate difference of the dual-comb (i.e., the fundamental limit of the frame rate) multiplied by the number of comb lines (i.e., the sampling ratio times the image resolution) defines the RF bandwidth $B_{RF}$ of the sampling frequencies, which must be less than half the repetition rate $f_{rep}$ of each optical frequency comb to avoid aliasing:
> $B_{RF} = \Delta f_{rep} \cdot M = f_{frame} \cdot \beta \cdot N^2  < \frac{f_{rep}}{2}$.
> Increasing the sampling ratio (more comb lines) reduces the frame rate, and vice versa. This fundamental constraint must be carefully balanced when designing dual-comb compressive imaging systems.”
>
> **Question 5: noise robustness claims** We are extending the noise study to multiple samples and reporting aggregate metrics. These results will be included in the final version.
>
> Lastly, we appreciate the reviewer’s valuable comments and suggestions. We believe that the combination of (i) the first dual-comb ghost imaging hardware demonstration and (ii) the Ghost-GPT dual-token transformer model provides substantive novelty. We will clarify dataset choices, add DL baselines, expand noise tests, and better explain system-level constraints to address the reviewer’s concerns.

---

> > ### Comment · Reviewer_wkCt · 2025-08-08
> > **Valuable Domain Contributions, but Limited ML Novelty — Rating Improved Post-Rebuttal**
> >
> > Dear Authors,
> >
> > Thank you for the effort in considering and replying to all the points mentioned in the review. I apologize for replying so late in the discussion phase. The rebuttal has improved my understanding of the contributions of the paper. Consider further clarifying some more aspects on the contributions.
> >
> > ### Regarding novelty and limited ML contributions:
> >
> > 1. Please help me understand what is the novelty in this — *"we combine speckle patterns and bucket sums as input tokens"*. For example, without this technique, how would this task (applying transformer to this data) be accomplished? And how does this combination lead to *"improved convergence and faster training"*?
> > Consider elaborating further on how this is different from a simple use of transformer architecture in place of a CNN/UNet architecture for this image-to-image regression task. This is the most important clarification needed.
> >
> > 2. *"adopting the dual-comb interferometry method for the first time in a ghost imaging experiment"* — even if this has not been done before, it is not an ML contribution. Further, if this is claimed as a contribution/novelty, the experiments have to be directed towards showing the performance improvement provided by this contribution. For example, how the image reconstruction improves by use of dual-comb over reconstruction that does not use dual-comb.
> > Thus, if this is indeed the main contribution of the paper, there is a mismatch between the contributions and experiments.
> > (More clarification on experiments requested in the next section "Regarding use of datasets".)
> >
> > ### Regarding use of datasets
> >
> > Thank you for your clarification about the datasets and the method used for capturing speckle patterns. The fact that this dataset is captured using the optical setup and not simulated mathematically should be clarified for an ML audience without a strong background in imaging. It is also a useful discussion on why currently ghost imaging experiments are generally conducted on binary images.
> >
> >
> > ### Limited diversity of baselines)
> >
> > Similar to the above discussion on claimed contributions, in this context, the claimed contributions of
> > - *"achieving video-rate imaging without dual-comb interferometry would be extremely challenging"*
> > - and that without a large number of uncorrelated speckle patterns, the reconstruction quality of other reconstruction algorithms will suffer (*"generating a large number of uncorrelated speckle patterns is not technically straightforward"*).
> >
> > need to be substantiated with corresponding experiments with and without the proposed contribution.
> >
> > ### Revisions/addition to the paper
> >
> > Multiple answers in the rebuttal were on the lines of adding major content to the paper (adding DL baselines, expanding noise tests, etc.). Such a major revision of the paper would be more suitable for resubmission, preventing a straight accept rating. However, based on the clarifications in the rebuttal, the overall rating of the paper can be increased.
> >
> >
> > ### Conclusions
> >
> > The demonstration of dual-comb ghost imaging hardware, enabling video-rate endoscopic imaging, and adopting the dual-comb for the first time in a ghost imaging experiment are valuable contributions but are not ML contributions. Also, as elaborated above, these need to be backed by corresponding experiments.
> >
> > However, based on the clarifications in the rebuttal, the overall rating of the paper can be increased.

---

> > > ### Author Response · Authors · 2025-08-09
> > > **Additional clarification (PART 1)**
> > >
> > > We thank the reviewer for the follow-up and for the opportunity to clarify.
> > >
> > > **Regarding novelty and limited ML contributions:**
> > >
> > > **RE: Comment 1**
> > >
> > > In Section 2 of our manuscript (“Background on Single-Pixel and Ghost Imaging”), we describe two main deep learning strategies used in ghost imaging:
> > >
> > > 1. Two-stage (classical + DL): A classical algorithm first reconstructs a low-quality image, followed by a neural network that performs denoising or super-resolution. This is analogous to conventional image upscaling pipelines.
> > >
> > > 2. End-to-end: The network takes bucket sums as input, directly outputting the reconstructed image, bypassing the classical step entirely. This yields faster reconstruction, which is critical for our dual-comb setup’s ultra-fast imaging rates.
> > >
> > > Prior end-to-end transformer works in ghost imaging (Ren, Wenhan, et al. "Ghost translation: an end-to-end ghost imaging approach based on the transformer network." Optics Express 30.26 (2022): 47921-47932.) were trained using solely bucket sums measurement, without the embedding of the per-measurement speckle patterns. In contrast, our Ghost-GPT model follows the end-to-end approach but introduces a key departure from prior work: we encode paired tokens that jointly contain the speckle pattern and its corresponding bucket sum. This allows the transformer to attend over both spatial structure (from the pattern) and measured intensity (from the bucket) in a unified representation. This speckle-aware model leverages more information during training, enabling better convergence and faster learning.
> > >
> > > To clarify, “the improved convergence and faster training” refers to the stability of training as well as the smaller number of epochs necessary for the model’s loss to asymptote. Training runs without speckle pattern embedding were less stable — they required more careful hyperparameter tuning and multiple attempts to obtain convergence in training— whereas training with the paired tokens converged more consistently in our setup. In addition, we observed the following in our experiments: the speckle patterns embedded into the tokens improves reconstruction fidelity (Fig. 7); specifically, as we increase the size of the speckle pattern embedding, the MSE decreases.
> > >
> > >
> > > **RE: Comment 2**
> > >
> > > We agree that the dual-comb interferometry method alone is not an ML contribution. However, our novelty lies in the integrated combination of dual-comb detection, optical fiber-based imaging, and advanced ML models. Simply put, our work exemplifies the co-design of imaging hardware and methodology (dual-comb ghost imaging) with advanced ML models, with the goal of achieving video-rate endoscopic imaging under extreme hardware constraints. The ultrafast detection enabled by the dual-comb source, together with the miniature imaging conduit provided by the optical fiber, addresses key bottlenecks that have restricted prior, and most importantly, useful implementations of ghost imaging.
> > >
> > > Most prior ML-only algorithmic works on ghost imaging assume that a large number of speckle patterns are measured simultaneously; however, this assumption is invalid because speckle data are typically acquired serially at a slow rate, making it impossible to image rapidly moving objects. From an ML contribution perspective, our work makes the assumption of a simultaneous imaging matrix (speckle pattern) valid and enables high-quality, simultaneous data acquisition even for rapidly moving objects in real-world settings. We would like to emphasize that good imaging requires both real-time capability and high fidelity, and our work focuses on achieving both—not just the high-fidelity aspect—through the co-design of the entire imaging system (hardware + software).
> > >
> > > These advances open the door to practical applications of ML-enhanced video-rate ghost imaging—such as endomicroscopy—and thereby enable new opportunities for machine learning methods tailored to this emerging domain and expand the compressed sensing field as whole. By removing longstanding experimental bottlenecks, our hardware advances in ghost imaging pave the way for applying more advanced, domain-specific, and novel ML methods.
> > >
> > > Regarding the reviewer’s comment, “Further, if this is claimed as a contribution/novelty, the experiments have to be directed towards showing the performance improvement provided by this contribution… there is a mismatch between the contributions and experiments,” we respectfully disagree. As discussed above, imaging speed—along with image fidelity—is a critical performance metric in real-life applications, and our dual-comb ghost imaging work has already demonstrated this speed improvement while maintaining good image fidelity.

---

> > > ### Author Response · Authors · 2025-08-09
> > > **Additional clarification (PART 2)**
> > >
> > > **Regarding use of datasets**
> > >
> > > **Thank you for your clarification about the datasets and the method used for capturing speckle patterns. The fact that this dataset is captured using the optical setup and not simulated mathematically should be clarified for an ML audience without a strong background in imaging. It is also a useful discussion on why currently ghost imaging experiments are generally conducted on binary images.**
> > >
> > > Thank you for your comments. We will clarify these points in the revised manuscript.
> > >
> > >
> > > **Limited diversity of baselines)**
> > >
> > > **Similar to the above discussion on claimed contributions, in this context, the claimed contributions of
> > > "achieving video-rate imaging without dual-comb interferometry would be extremely challenging"
> > > and that without a large number of uncorrelated speckle patterns, the reconstruction quality of other reconstruction algorithms will suffer ("generating a large number of uncorrelated speckle patterns is not technically straightforward").
> > > need to be substantiated with corresponding experiments with and without the proposed contribution.**
> > >
> > > We respectfully disagree that a new, detailed experimental optics benchmark with a traditional ghost imaging setup is necessary for this work. Such an experiment would require building a separate optical system, representing a substantial resource commitment, and repeating experiments of well-established, prior works to compare with our experiment is beyond the scope of our paper. Moreover, the speed and size limitations of conventional ghost imaging setups using spatial light modulators (SLMs) or digital micromirror devices (DMDs) have already been extensively documented in the cited literature, and our work directly addresses these known bottlenecks through the co-designed hardware–ML approach we propose.
> > >
> > >
> > > **Revisions/addition to the paper**
> > >
> > > **Multiple answers in the rebuttal were on the lines of adding major content to the paper (adding DL baselines, expanding noise tests, etc.). Such a major revision of the paper would be more suitable for resubmission, preventing a straight accept rating. However, based on the clarifications in the rebuttal, the overall rating of the paper can be increased.**
> > >
> > >
> > > Thank you for your comments. Most of the original remarks and criticisms stem from a misunderstanding of the novelty of our work and the limitations imposed by the experimental constraints (specifically, the limited number of speckle patterns). We will clarify these points, which can be easily misunderstood by the software-oriented ML community. Other requests from the reviewers (e.g., adding DL baselines, expanding noise tests) can be addressed relatively easily; however, we believe these do not constitute major changes in light of our clarified novelty and the constraints of the experimental setup. For example, while adding comparisons with other DL baselines is a reasonable addition, as we discussed in the rebuttal, such comparisons are of limited value here because the performance improvements are incremental due to the use of simple binary datasets.
> > >
> > >
> > > **Conclusions**
> > >
> > > **The demonstration of dual-comb ghost imaging hardware, enabling video-rate endoscopic imaging, and adopting the dual-comb for the first time in a ghost imaging experiment are valuable contributions but are not ML contributions. Also, as elaborated above, these need to be backed by corresponding experiments.
> > > However, based on the clarifications in the rebuttal, the overall rating of the paper can be increased.**
> > >
> > > Thank you for your encouraging comments. Although our major contribution may be viewed as non-ML, our work focuses on the co-design of hardware and software for ML-enhanced, video-rate ghost imaging. Given the increasing importance of high-quality data in ML research as well as the need to explore novel domains which can take advantage of ML techniques, we believe our contribution should also be regarded as an ML contribution. Importantly, since our paper was submitted to the “Applications (e.g., vision, language, speech and audio, Creative AI)” track, we believe that application-specific or application-aware co-design of hardware and software is valuable, as opposed to software-only ML contributions.
> > >
> > > Also, as explained above, we have also outlined why additional experiments are not necessary to substantiate our novelty and contribution.
> > >
> > > Again, thank you very much for the invaluable comments and fruitful discussion.

---

### Note · Authors · 2025-08-13

Our work demonstrates high-fidelity, video-rate computational ghost imaging using only a single-core fiber and a single-pixel detector by combining dual-comb interferometry with a Transformer tailored for fast, high-quality reconstruction—a hardware–software co-design.

Some initial reviews emphasized only the ML component and overlooked the dual-comb experiment and the constraints that shaped our model. In this first dual-comb ghost-imaging prototype, (i) we used binary datasets because the number of uncorrelated speckle patterns was limited, and (ii) we compared against classical algorithms rather than other ML models because, with such simple datasets, the MSE improvement over other ML models was only incremental. Our work is the first experimental demonstration of dual-comb ghost imaging, and the concerns in the initial reviews including the absence of realistic grayscale reconstructions reflect prototype-stage constraints, not a limitation of the method. We’ve discussed this in detail in our rebuttal. Also, our Transformer-base model is application-aware: paired tokens jointly encode each speckle pattern and its corresponding bucket sum to meet real-time, high-fidelity goals.

Much prior ML-centric work in computational ghost imaging optimizes reconstruction quality and speed of the ML-model itself while assuming many speckle patterns can be measured simultaneously, paying little attention to how patterns are generated or objects detected in a real imaging system. In practice, SLM/DMD systems acquire patterns serially at low rates, making fast-motion imaging infeasible. Our co-designed system makes the “simultaneous imaging matrix” assumption valid by enabling high-quality, simultaneous acquisition—even for rapidly moving objects in real-world settings.

Given the central role of data quality in ML and the value of application-driven co-design, we believe this contribution is significant for the ML community. Whereas prior ML-only approaches largely target stationary scenes, our method acquires high-quality data for moving objects as well, expanding the practical scope of ML-based computational ghost imaging. Aligned with the Applications track, our results demonstrate that hardware–software co-design enables capabilities—specifically, video-rate ghost imaging for endomicroscopy with minimal hardware—that software-only ML cannot achieve.

We hope these remarks clarify the novelty and significance of our work. Thank you for your consideration.

---

### Decision · Program_Chairs · 2025-09-17

**Decision:**

Accept (poster)

**Comment:**

After the rebuttal, the reviewers were mostly satisfied with the authors’ responses and upgraded their ratings.
The paper ultimately received three acceptances and one borderline rejection.
As AC, I support the recommendation for acceptance.
The authors are encouraged to incorporate their rebuttal responses into the camera-ready version.